

# Nonequilibrium quasiparticle distribution in superconducting resonators: Effect of pair-breaking photons

**Paul B. Fischer[1,2] and Gianluigi Catelani[1,3]**

**1** JARA Institute for Quantum Information (PGI-11),
Forschungszentrum Jülich, 52425 Jülich, Germany
**2** JARA Institute for Quantum Information,
RWTH Aachen University, 52056 Aachen, Germany
**3** Quantum Research Center, Technology Innovation Institute,
Abu Dhabi 9639, UAE

## Abstract

Many superconducting devices rely on the finite gap in the excitation spectrum of a superconductor: thanks to this gap, at temperatures much smaller than the critical one the number of excitations (quasiparticles) that can impact the device's behavior is exponentially small. Nevertheless, experiments at low temperature usually find a finite, non-negligible density of quasiparticles whose origin has been attributed to various nonequilibrium phenomena. Here, we investigate the role of photons with energy exceeding the pair-breaking threshold $2\Delta$ as a possible source for these quasiparticles in superconducting resonators. Modeling the interacting system of quasiparticles, phonons, sub-gap and pair-breaking photons using a kinetic equation approach, we find analytical expressions for the quasiparticles' density and their energy distribution. Applying our theory to measurements of quality factor as function of temperature and for various readout powers, we find they could be explained by assuming a small number of photons above the pair-breaking threshold. We also show that frequency shift data can give evidence of quasiparticle heating.


doi:10.21468/SciPostPhys.17.3.070

# 1   Introduction

A detector should ideally respond in a well-understood way to the physical quantity under investigation while being unaffected by other processes. However, unwanted effects can contribute to noise that obscures the signal. The noise mechanism can be specific to the type of detector: even within superconducting detectors, different mechanisms are prominent for different designs. For example, in superconducting nanowire single photon detectors [1, 2] vortices crossing the nanowire lead to dark counts [3, 4], and in kinetic inductance detectors (KIDs) [5] the generation-recombination noise due to the creation and annihilation of quasiparticles increases the noise equivalent power [6–8]. A KID consists of a resonator whose inductance and hence resonant frequency changes when photons of energy above twice the superconducting gap $\Delta$ break Cooper pairs. The response of the resonator is monitored via a probe tone that maintains a large number $\bar{n} \gg 1$ of photons with frequency $\omega_0 < 2\Delta$ in the resonator (hereinafter we set $\hbar = k_B = 1$). The device is operated at temperatures much smaller than the critical one, $T \ll T_c$, where the thermal equilibrium number of quasiparticles is expected to be negligible. However, a variety of experiments with not only resonators [9, 10] but also superconducting qubits [11, 12] indicates the presence of many quasiparticles whose origin is often unclear. These excess quasiparticles influence basic parameters of the resonator such as the quality factor, which is why the goal of this work is to quantify their effect and propose a possible source.

To model the non-equilibrium state of a superconductor, a system of coupled kinetic equations for the distribution functions of quasiparticles and phonons has been proposed long ago [13]; even in the steady-state and for uniform systems – that is, considering only the energy dependence of the distribution functions, – these equations are usually solved numerically [14, 15]. Only recently, approximate analytical solutions have been obtained in the parameter regime relevant to KIDs [16, 17]. The results of these two works can be summarized as follows: a large number of (non-pair-breaking) photons can "heat up" the quasiparticles by pushing them to higher energy as compared to the phonon temperature; these quasiparticles can relax by emitting phonons, so that the latter are also driven out of equilibrium. Quasiparticles pushed to energy above $3\Delta$ can emit a relatively large number(as compared to the thermal equilibrium value of pair-breaking phonons with frequency $\omega > 2\Delta$; because of these phonons, in the steady state the quasiparticle density is then much larger than the equilibrium value, a situation that is encountered at low temperatures. These findings can quantitatively describe the experimental data for a resonator's internal quality factor $Q_i$ of Ref. [18] at intermediate temperatures at which the number of quasiparticles is close to the equilibrium value but their distribution function is not thermal. As temperature decreases, $Q_i$ is predicted to



Figure 1: A full model of a thin-film superconducting resonator (dark box) takes into account the resonant mode of frequency $\omega_0$, quasiparticles, phonons, and their interactions (with coupling constants $c_{Phot}^{QP}$ and $\tau_0$). Additionally, quasiparticles can be generated by pair-breaking photons of frequency $\omega_{PB}$ (coupling constant $c_{Phot,PB}^{QP}$; this process was not considered in Ref. [17]) and the phonons interact with the substrate, which act as a thermal bath (coupling constant $\tau_l$).

saturate, but at values few to several orders of magnitude larger than those measured.

The decrease of $Q_i$ with readout power observed at low temperature in Ref. [18] is incompatible with the expectation from losses due to two-level systems [19]. On the other hand, clear evidence for the presence of pair-breaking photons is provided by measurements of so-called parity switching rates in superconducting qubits: effects initially attributed to "hot" quasiparticles [20] have been explained in terms of tunneling assisted by pair-breaking photons [21], as confirmed by additional experiments [22,23]. Motivated in part by these results, here we extend the model [16,17] discussed in the previous paragraph to include the effect of a small number of *pair-breaking* photons of frequency $\omega_{PB} > 2\Delta$ and show that they could be responsible for the low-temperature behavior of $Q_i$ reported in Ref. [18].

In Sec. 2 we present the kinetic equation that determines the quasiparticle distribution; it extends the previously used kinetic equations describing the interaction of quasiparticles with photons of energy below the pair breaking threshold $2\Delta$ [17,24,25] by including a contribution from a mode of energy above the threshold. In Sec. 3 we derive approximate analytical solutions for the case of zero temperature and negligible number of photons below the threshold, and validate the results numerically. The effect of the low-energy photons on the distribution's shape is investigated in Sec. 4. In Sec. 5 the results of the preceding section are used to calculate quality factor and resonance frequency shift in thin-film resonators, and we analyse experimental data [18] for this quantities. Section 6 summarizes our findings.

## 2 Kinetic equation

The kinetic equation for the quasiparticle distribution function $f(E)$ in a homogeneous superconductor has the form

$$\frac{df(E)}{dt} = St^{Phon}\{f,n\} + St^{Phot}\{f,\bar{n}\} + St_{PB}^{Phot}\{f,\bar{n}_{PB}\}, \tag{1}$$

with $E$ the energy measured from the Fermi level. The collision integrals in the right-hand side account for the interaction between quasiparticles and phonons, $St^{Phon}\{f,n\}$, non-pair-breaking photons, $St^{Phot}\{f,\bar{n}\}$, and pair-breaking ones, $St_{PB}^{Phot}\{f,\bar{n}_{PB}\}$, respectively. While this structure of the kinetic equation is quite general, we will mostly consider thin-film superconducting resonators, as depicted in Fig. 1. The phonon collision integral $St^{Phon}\{f,n\}$ and the photon one for non-pair breaking photons $St^{Phot}\{f,\bar{n}\}$ can be found for instance in Ref. [17]. In this work we extend the model by including photons of energy $\omega_{PB} > 2\Delta$ via the collision integral:

$$St_{PB}^{Phot}\{f,\bar{n}_{PB}\} = St^{Phot}\{f,\bar{n}_{PB}\} + St_{PB,g}^{Phot}\{f,\bar{n}_{PB}\} + St_{PB,r}^{Phot}\{f,\bar{n}_{PB}\}. \tag{2}$$

Here, $St^{Phot}\{f, \bar{n}_{PB}\}$ is a number-conserving scattering term accounting for the redistribution of quasiparticles in energy due to the absorption of pair-breaking photons. It can be obtained from the photon integral $St^{Phot}$ for non-pair-breaking photons of frequency $\omega_0$ [see diagrams a) and b) in Fig. 2],

$$St^{Phot}\{f, \bar{n}\} = c_{Phot}^{QP} U^+(E, E + \omega_0) \Big\{ f(E + \omega_0)[1 - f(E)](\bar{n} + 1) - f(E)[1 - f(E + \omega_0)]\bar{n} \Big\}$$
$$+ c_{Phot}^{QP} U^+(E, E - \omega_0) \Big\{ f(E - \omega_0)[1 - f(E)]\bar{n} - f(E)[1 - f(E - \omega_0)](\bar{n} + 1) \Big\}, \tag{3}$$

by replacing photon energy $\omega_0$, photon number $\bar{n}$ and coupling constant $c_{Phot}^{QP}$ by the respective values of the pair breaking photons $\omega_{PB}$, $\bar{n}_{PB}$ and $c_{Phot,PB}^{QP}$ [the appearance of $\bar{n}$ or $\bar{n}_{PB}$ in the argument of teh collision integral indicates which of these quantities should be used]. We use here the notation of Ref. [17] by defining $U^{\pm}(E_1, E_2) = K^{\pm}(E_1, E_2)\rho(E_2)$, with BCS coherence factor $K^{\pm}(E_1, E_2) = 1 \pm \Delta^2/(E_1 E_2)$ and BCS density of states $\rho(E_2) = E_2/\sqrt{E_2^2 - \Delta^2}$ (see *e.g.* Ref. [26] for a discussion of coherence factors). This collision integral conserves the number of quasiparticles, $\int_\Delta dE \, \rho(E) St^{Phot} = 0$. We note that both for non- and pair-breaking photons we consider a single mode of definite frequency; this is justified for non-pair-breaking photons by the fact that, as mentioned in the Introduction, in applications such as KIDs one high-quality factor mode of the resonator is probed. For pair-breaking photons this is in general a simplification; however, at least in some of the regimes we will consider, the extension to multiple modes is straightforward (see Secs. 3 and 4). Moreover, we will show that this simplification does not alter qualitatively our interpretation of experimental data for the quality factor in Sec. 5.

In addition to the scattering term, there is a term accounting for the generation of new quasiparticles [diagram d) in Fig. 2],

$$St_{PB,g}^{Phot} = c_{Phot,PB}^{QP} U^-(E, \omega_{PB} - E)\bar{n}_{PB}[1 - f(E)][1 - f(\omega_{PB} - E)], \tag{4}$$

and a term describing recombination accompanied by the emission of a photon,

$$St_{PB,r}^{Phot}\{f, \bar{n}_{PB}\} = -c_{Phot,PB}^{QP} U^-(E, \omega_{PB} - E)(1 + \bar{n}_{PB})f(\omega_{PB} - E)f(E). \tag{5}$$

In Appendix A we discuss how to estimate the coupling constants $c_{Phot}^{QP}$ and $c_{Phot,PB}^{QP}$. The phonon collision integral can be obtained from the pair-breaking photon one, Eq (2), by replacing $U^{\pm} \to U^{\mp}$, $\omega_{PB} \to \omega$, $c_{Phot,PB}^{QP} \to \omega^2/\tau_0 T_c^3$, $\bar{n}_{PB} \to n(\omega)$, and integrating over $\omega > 0$; the integration limits are chosen so that the second argument of $U^{\pm}$ is larger than $\Delta$. In these replacements, $n(\omega)$ is the phonon distribution function and $\tau_0$ is the time scale characterizing the strength of the electron-phonon interaction [27].

To complete the description of the system, one should also consider the kinetic equation for the phonon distribution function $n(\omega)$; we don't give it here as it will not be needed explicitly, and rather refer the reader to our previous work [17]. That equation contains a thermalization term $-[n(\omega) - n_T(\omega, T_B)]/\tau_l$ accounting for the relaxation of the phonons back to equilibrium at temperature $T_B$ over the time scale $\tau_l$. For the numerical calculations in this work, we either assume fast equilibration to zero temperature, $n(\omega) = 0$, or solve the full coupled system of kinetic equations for quasiparticle and phonons, as will be specified. The numerical solutions are obtained using a straightforward extension of the algorithm described in Ref. [17].

In the next sections, we discuss approximate analytical solutions to Eq. (1) in the steady state $df/dt = 0$. While different approximations will be employed in different regimes, some approximations are common to all cases we will consider: we generally assume small occupation probability at all energies, $f \ll 1$, so that we can replace Pauli blocking factors with unity,

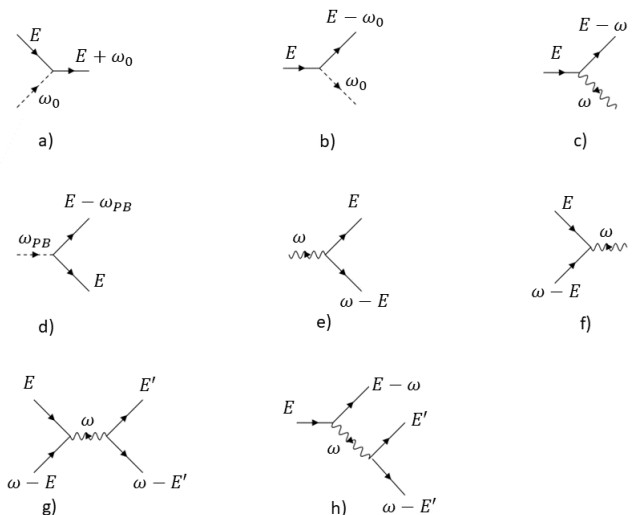

Figure 2: Schematic depiction of the main processes entering the collision integrals in Eq. (1). Straight lines correspond to quasiparticles, wavy lines to phonons, and dashed lines to photons. Diagrams a) and b) represent the absorption and emission of non-pair-breaking photons [cf. Eq. (3)], c) the emission of a phonon [c.f. first term in the bracket of Eq. (6)], d) quasiparticle generation by pair-breaking photons [Eq. (4)], a process not included in Ref. [17]. Diagram e) depicts generation by a pair-breaking phonon [last term of Eq. (6)], f) provides the bare recombination rate $r_0 x_{qp}$, and g) renormalizes the bare recombination coefficient $r_0$ to $r$ in Eq.(6) [see also Eq. (8)]. Diagram h) describes an additional pair-breaking mechanism due to phonon emitted by quasiparticles with energy $E > 3\Delta$; this process is not included in the low-energy approximation of Eq. (6), but if $\bar{n}$ is sufficiently large (cf. Sec. 4) it can affect the density, see the term proportional to $G(T_*/\Delta)$ in Eq. (49) and Ref. [17]. In Sec.. 3 we only consider c), d) and f), in Sec. 4 we include diagrams a) and b), and in Secs. 4.3 and 5 we consider all diagrams. We do not include the diagram corresponding to photon mediated recombination [similar in structure to f)], as its contribution is generally negligible compared to phonon mediated recombination (see Sec. 3.1).

$(1-f) \to 1$. The validity of this assumption, which limits our considerations to temperatures small compared to the critical one and small number of pair-breaking photons, can be checked once the solution to the kinetic equation is found; importantly, number-conserving contributions to the collision integrals [cf. Eq (3)] remain number conserving in this approximation. Also, we focus on quasiparticles of low energy, $\Delta < E \lesssim 2\Delta$; assuming the phonon bath temperature $T_B$ to be sufficiently low (at least compared to the gap, although more stringent conditions will be discussed), phonon absorption can be neglected and the phonon collision integral is approximately given by [cf. Ref. [16] and diagrams c), e), and f) in Fig. 2]

$$St^{Phon}\{f,n\} \simeq \frac{1}{\tau_0 T_c^3} \int_0^\infty d\omega\, \omega^2 \frac{2\epsilon + \omega}{\sqrt{2\Delta(\epsilon+\omega)}} f(\epsilon+\omega)$$
$$-\left[\frac{128}{105\sqrt{2}}\left(\frac{\Delta}{T_c}\right)^3\left(\frac{\epsilon}{\Delta}\right)^{7/2}\frac{1}{\tau_0} + r x_{qp}\right]f(\epsilon) + St_g^{Phon}, \tag{6}$$

where $\epsilon = E-\Delta$ is the energy measured from the gap [with a slight abuse of notation, we also substitute $f(E) = f(\Delta+\epsilon) \to f(\epsilon)$] and $St_g^{Phon}$ is obtained from Eq. (4) following the steps

given after Eq. (5). The last term inside square brackets accounts for quasiparticle recombination accompanied by phonon emission; here we have introduced the quasiparticle density normalized by the Cooper pair density, $x_{qp} = N_{qp}/2\rho_F\Delta$, where the quasiparticle density is given by

$$N_{qp} = 4\rho_F \int_\Delta dE\, \rho(E)f(E). \qquad (7)$$

The parameter $r = 4(\Delta/T_c)^3/\bar{\tau}_0$ is the (normalized) recombination coefficient, where

$$\bar{\tau}_0 = \tau_0(1 + \tau_l/\tau_0^{PB}), \qquad (8)$$

depends on the thermalization time $\tau_l$ of the phonons and $\tau_0^{PB}$ is the lifetime of a phonon at the pair-breaking threshold [the reduction of the recombination coefficient for $\tau_l \neq 0$ accounts for the processes in diagram g)]. This time is proportional to $\tau_0$ times the ratio between ion density over Cooper pair density and times $(T_c/\omega_D)^3$, with $\omega_D$ the Debye frequency; for aluminum, we use $\tau_0/\tau_0^{PB} = 1.7 \times 10^3$ [17]. The two terms inside square bracket in Eq. (6) define the energy-dependent spontaneous phonon emission rate

$$\frac{1}{\tau_{e,n}^{qp}(\epsilon)} = \frac{128}{105\sqrt{2}}\left(\frac{\Delta}{T_c}\right)^3\left(\frac{\epsilon}{\Delta}\right)^{7/2}\frac{1}{\tau_0}, \qquad (9)$$

and the density-dependent phonon-assisted recombination rate

$$\frac{1}{\tau_r^{qp}} = rx_{qp} = RN_{qp}, \qquad (10)$$

with the recombination coefficient $R = 2\Delta^2/\rho_F\bar{\tau}_0T_c^3$ (which, strictly speaking, depends weakly on energy and/or the width of the distribution function above the gap – that is, the "effective" quasiparticle temperature, – see [17, 27]; we neglect this factor-of-order-one correction in this work). The fact that the recombination time depends on the density represents the main non-linearity of the kinetic equation; as we will show, however, the density can be calculated without a detailed solution of the kinetic equation in an approach reminiscent of the phenomenological one pioneered by Rothwarf and Taylor [28]. In fact, to use that approach one must also take into account the last term in the right-hand side of Eq. (6), namely the generation of quasiparticles by phonons; for $f \ll 1$, that term depends only on the phonon distribution function $n$ and not on $f$. In this work we will not need the detailed form of $St_g^{Phon}$ (see e.g. Ref. [17]), rather the integral of its product time the quasiparticle density of states; for phonons in thermal equilibrium at temperature $T_B$ we have

$$\int dE\,\rho(E)St_g^{Phon} = r_0\pi T_B e^{-2\Delta/T_B} \equiv r_0 G_{T_B}/4r\rho_F, \qquad (11)$$

where $r_0$ is the recombination coefficient $r$ for $\tau_l = 0$ and in the last term we introduce the notation for the generation coefficient $G_{T_B}$ at the given temperature $T_B$.

Two additional approximations can be introduced for the collision integral with pair-breaking photons, Eq. (2). First, for quasiparticles in the relevant energy range, in $St^{Phot}\{f, \bar{n}_{PB}\}$ the terms proportional to $U^+(E, E-\omega_{PB})$ [cf. Eq. (3)] vanish since $E - \omega_{PB} < \Delta$; we also assume that $f \ll 1$ at all energies and that the distribution function decays quickly with increasing energy, so that $f(E + \omega_{PB}) \ll f(E)\bar{n}_{PB}/(\bar{n}_{PB} + 1)$, and arrive at

$$St^{Phot}\{f, \bar{n}_{PB}\} \simeq -c_{Phot,PB}^{QP}U^+(E, E+\omega_{PB})\bar{n}_{PB}f(E)$$

$$\simeq -c_{Phot,PB}^{QP}\bar{n}_{PB}f(E) = -\frac{1}{\tau_{abs,PB}^{qp}}f(E), \qquad (12)$$

where in the last equality we have introduced the quasiparticle lifetime against the absorption of pair-breaking photon. Second, we assume that photon-assisted recombination can be neglected in comparison to the phonon-assisted one, that is [see Eqs. (5) and (10)]

$$c_{Phot,PB}^{QP} U^-(E, \omega_{PB} - E)(1 + \bar{n}_{PB})f(\omega_{PB} - E) \ll r x_{qp}. \tag{13}$$

Clearly, this inequality is violated as $E \to \omega_{PB} - \Delta$ due to the divergence of the superconducting density of states; however, we will show that the violation happens so extremely close to that energy to have no impact on our results.

In closing this introductory section, we note that in all the formulas above the gap value $\Delta$ should be understood as given by the self-consistent equation for the order parameter. In the presence of a small density of quasiparticles, the gap is smaller than its no-quasiparticle value $\Delta_0$; defining $\delta\Delta = \Delta_0 - \Delta$, at leading order we have $\delta\Delta/\Delta_0 \simeq x_{qp}$ [17] and for most purposes the difference between $\Delta$ and $\Delta_0$ can be ignored. One exception is the evaluation of the resonator's frequency shift, which we discuss in Sec. 5.1.

# 3   Generation by pair breaking photons

As a first step, in this Section we study the steady-state quasiparticle distribution in the presence of pair-breaking photons only, a situation complementary to that analyzed in Refs. [16,17] in which only non-pair-breaking photons were taken into account. Concretely, we assume that there are no modes below the pair-breaking threshold and we set $St^{Phot}\{f, \bar{n}\} = 0$ (that is, $c_{Phot}^{QP} = 0$). We also assume that the phonons are at zero temperature, $n(\omega) = 0$, or in other words fast thermalization ($\tau_l \to 0$) with a $T_B = 0$ bath. Therefore, the kinetic equation reduces to

$$0 = St^{Phot}\{f, \bar{n}_{PB}\} + St_{PB,g}^{Phot}\{f, \bar{n}_{PB}\} + St^{Phon}\{f, n\}, \tag{14}$$

with the pair-breaking photon collision integrals of Eqs. (3) [appropriately modified for pair-breaking photons] and (4) and the phonon collision integral of Eq. (6) [in which by assumption the last term accounting for generation by phonons vanishes].

Even in this simplified case, we can in principle distinguish two different regimes, depending on which process is dominant near the gap. Since at those energies phonon-assisted recombination is faster than phonon emission [$1/\tau_{e,n}^{qp}(\epsilon) \ll 1/\tau_r^{qp}$ as $\epsilon \to 0$], we should compare the rate for the former process, $1/\tau_r^{qp}$ [Eq. (10)], to the rate of absorption of photons, $1/\tau_{abs,PB}^{qp}$ [Eq. (12)]. We show below (Sec. 3.1) that the situation of experimental relevance for Al resonators is

$$\frac{1}{\tau_{abs,PB}^{qp}} \ll \frac{1}{\tau_r^{qp}}. \tag{15}$$

In fact, to check when this inequality holds, we need to eliminate the dependence of the right-hand side on the quasiparticle density. To that end, we multiply Eq. (14) by $\rho(E)$, integrate over $E$, and use that $St^{Phot}$ is number-conserving to find, for $f(\epsilon) \ll 1$,

$$rx_{qp}^2 = \frac{2}{\Delta} \int\limits_{\Delta}^{\omega_{PB}-\Delta} dE\, \rho(E) St_{PB,g}^{Phot}$$

$$= 2c_{Phot,PB}^{QP} \bar{n}_{PB} S_-(2 + \xi/\Delta), \tag{16}$$

where

$$\xi = \omega_{PB} - 2\Delta, \tag{17}$$

and the structure factor $S_-(x)$ [21, 29] has the approximate form $S_-(x) \simeq \pi(x-2)/2$ for $x - 2 \ll 2$. Using Eq. (16), the inequality in Eq. (15) can be rewritten as

$$c_{Phot,PB}^{QP} \bar{n}_{PB} \ll \pi r \xi / \Delta, \tag{18}$$

where from now on we assume $\xi < \Delta$ ($\omega_{PB} < 3\Delta$). When the above condition is true, in most cases quasiparticles created by photon pair breaking relax towards the gap and/or recombine by phonon emission before they can absorb a photon; indeed, the condition can be satisfied if the number of photon is sufficiently small. If a photon is absorbed before a recombination event, this will lead to a second peak in the quasiparticle distribution function at energies above $3\Delta$. At those energies, quasiparticle relaxation by phonon emission is much faster than near the gap, so assuming that to be the dominant process, one can treat the second peak in a perturbative way, similarly to the "cold" regime of Ref. [16]. We do not pursue this approach further in this section, since if the condition in Eq. (15), or equivalently Eq. (18), is satisfied, the effect of the second peak can be neglected and only energies $\epsilon < \xi$ are relevant (see, however, Sec. 4.2 for the expression for the second peak).

So long as Eq. (15) holds, we can further simplify Eq. (14) by ignoring the first term on the right-hand side to get

$$St_{PB,g}^{Phot}\{f, \bar{n}_{PB}\} = -St^{Phon}\{f, n\}. \tag{19}$$

Next, we introduce the dimensionless energy variable

$$\gamma = \epsilon / \xi, \tag{20}$$

and the function

$$\phi(\gamma) = \sqrt{\frac{2\Delta}{\xi}} \frac{r x_{qp}}{c_{QP,PB}^{Phot} \bar{n}_{PB}} \left[ \left( \frac{\gamma}{\gamma_*'} \right)^{7/2} + 1 \right] f(\xi\gamma), \tag{21}$$

where the dimensionless parameter

$$\gamma_*' = \left( \frac{\tau_{e,n}^{qp}(\xi)}{\tau_r^{qp}} \right)^{2/7} = \left( \frac{105\sqrt{2} \, x_{qp}}{32} \right)^{2/7} \frac{\Delta}{\xi}, \tag{22}$$

determines whether at the highest energy at which quasiparticles are generated (that is, $\epsilon = \xi$) recombination is faster, $\gamma_*' > 1$, or slower, $\gamma_*' < 1$, than relaxation by phonon emission. With this notation, the steady-state equation (19) takes the form of a Volterra integral equation of the second kind

$$\frac{1}{\sqrt{1-\gamma}} = \phi(\gamma) - \int_\gamma^1 d\gamma' I(\gamma, \gamma') \phi(\gamma'), \tag{23}$$

with the kernel

$$I(\gamma, \gamma') = \frac{105}{128} \frac{\gamma + \gamma'}{\sqrt{\gamma'}} \frac{(\gamma - \gamma')^2}{(\gamma_*')^{7/2} + (\gamma')^{7/2}}. \tag{24}$$

In Eq. (23) the first term on the right-hand side accounts for quasiparticle out-scattering by spontaneous phonon emission and the second one for the corresponding in-scattering process; in that term the upper limit of integration is set to 1 consistently with ignoring absorption of pair-breaking photons and thus occupation at energies $\epsilon > \xi$, as discussed earlier in this section. The term on the left-hand sides originates from the quasiparticle generation by photons; its divergence as $\gamma \to 1$ originates from the diverging density of states and, as we will see, leads to a divergent distribution function. This unphysical divergence is in fact cut off by the Pauli blocking factors that we have neglected by assuming $f \ll 1$; however, the divergence is

integrable and to our knowledge it does not lead to unphysical behavior of any observable, so it is not further considered.

Depending on the value of $\gamma'_*$ we can distinguish two cases. For $\gamma'_* \gtrsim 1$, the solution to Eq. (23) can be given in terms of a Neumann series [30]

$$\phi(\gamma) = \sum_{j=0}^{\infty} \phi^{(j)}(\gamma), \tag{25}$$

with $\phi^{(0)}(\gamma) = 1/\sqrt{1-\gamma}$ and

$$\phi^{(j)}(\gamma) = \int_{\gamma}^{1} d\gamma' I(\gamma, \gamma') \phi^{(j-1)}(\gamma'), \tag{26}$$

over the whole range $\gamma \in [0, 1]$. In fact for $\gamma'_* \gg 1$ we can ignore $(\gamma')^{7/2}$ in the denominator of the kernel, Eq. (24), and readily see that the terms in the series are suppressed by $(\gamma'_*)^{-7j/2}$.

The second case, $\gamma'_* \ll 1$, is relevant to small quasiparticle density. In this case the solution has to be constructed differently. For $\gamma \gg \gamma'_*$, the kernel can be simplified to

$$I(\gamma, \gamma') \simeq \tilde{I}(\gamma, \gamma') \equiv \frac{105(\gamma' - \gamma)^2(\gamma + \gamma')}{128(\gamma')^4}. \tag{27}$$

Using this simplified kernel, the solution to Eq. (23) can again be written as a Neumann series, Eq. (25), but using the approximate kernel $\tilde{I}$ in Eq. (26) instead of $I$. The first and second order terms can be calculated analytically and are given in Appendix B. In fact, such a solution is valid under a weaker assumption than $\gamma \gg \gamma'_*$: Using the simplified kernel is a good approximation if the dominant contribution to the integral in Eq. (23) comes from the interval $\gamma' \in [\gamma'_*, 1]$. This holds for $\gamma \gg \gamma_*$, with $\gamma_*$ defined by

$$\int_{\gamma'_*}^{1} d\gamma' \phi(\gamma') \tilde{I}(\gamma_*, \gamma') = \int_{\gamma_*}^{\gamma'_*} d\gamma' \phi(\gamma') \tilde{I}(\gamma_*, \gamma') \tag{28}$$

(using the simplified kernel in the left-hand side is a reasonable approximation, while using it in the right-hand side overestimates the value of that integral, since the full kernel is smaller than the simplified one for small $\gamma'$; this leads to a conservative estimate for $\gamma_*$). While $\gamma'_*$ determines whether out-scattering at energy $\gamma$ is dominated by spontaneous phonon emission ($\gamma > \gamma'_*$) or recombination ($\gamma < \gamma'_*$), $\gamma_*$ distinguish whether in-scattering mostly originates from states for which spontaneous phonon emission ($\gamma > \gamma_*$) or recombination dominates ($\gamma < \gamma_*$). Equation (28) can be solved numerically using a bisection algorithm, the solution is presented in Fig. 3. We see there that $\gamma_*$ is at least an order of magnitude smaller than $\gamma'_*$; taking the geometrical average between the two quantities, we then estimate that the Neumann series constructed with the simplified kernel is a good approximation for $\gamma \gtrsim \gamma_c \equiv \gamma'_*/3$.

Except for the zeroth order, the terms in the "simplified" Neumann series diverge as $\gamma \to 0$; this is a result of the use of the simplified kernel, which has a non-integrable singularity for $\gamma = 0$ as $\gamma' \to 0$, while the exact kernel approaches zero in that limit. Thus, the solution for $\gamma \lesssim \gamma_c$ must be constructed differently. We proceed by considering a series expansion for $\phi$ up to third order in $\gamma$,

$$\phi(\gamma) = a_0 + a_1 \gamma + a_2 \gamma^2 + a_3 \gamma^3. \tag{29}$$

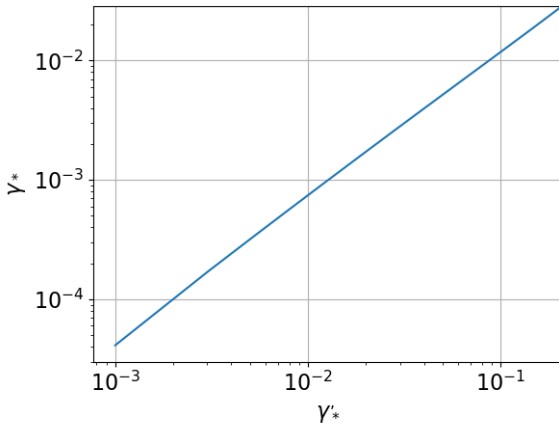

Figure 3: Numerically calculated $\gamma_*$ as function of $\gamma'_*$, determining the range of validity $\gamma > \gamma_*$ for the validity of the Neumann series constructed using the simplified kernel in Eq. (27).

Neglecting small factors of order $(\gamma_c/\gamma'_*)^{7/2}$ and higher, the lower integration limit in the right-hand side of Eq. (23) can be set to zero, and for consistency the left-hand side should be expanded also only up to third order in $\gamma$. Then the expansion coefficients for $\phi$ are given by

$$
\begin{aligned}
a_0 &= 1 + I_0\,, \\
a_1 &= \frac{1}{2} - I_1\,, \\
a_2 &= \frac{3}{8} - I_2\,, \\
a_3 &= \frac{5}{16} + I_3\,,
\end{aligned}
\tag{30}
$$

with

$$
I_l = \frac{105}{128} \int\limits_0^1 d\gamma' \phi(\gamma') \frac{\gamma'^{5/2-l}}{\gamma'^{7/2} + \gamma'^{7/2}_*}\,.
\tag{31}
$$

In this expression $\phi$ should be understood as the exact solution, extending over the whole interval $[0,1]$. The coefficients can be then calculated numerically, see the dots in Fig. 4. However, for an approximate estimate of the coefficients, we can set the lower integration limit in Eq. (31) to $\gamma_c$ and use the "simplified" Neumann series solution for $\phi$ including terms up to $j = 2$ (see Appendix B). The approximate coefficients calculated in this way are shown as lines; except for $a_3$, we find good agreement between exact and approximate coefficients. For later use, we note that $a_0 \simeq 0.073 \log^3(12/\gamma'_*) - 0.35 \log^2(12/\gamma'_*) + 1.23 \log(12/\gamma'_*)$ is a good approximation in the range covered by the dots in Fig. 4 (the dependence on powers of the logarithm originates from the behavior of $\phi^{(j)}$, $j = 0, 1, 2$, at small $\gamma$, while the numerical coefficients are obtained by comparison to the numerics).

We exemplify the results obtained in this section in Fig. 5. The black lines show the distribution function as obtained by numerical solution of the kinetic equation, while the colored lines are the analytical approximations. Both the "simplified" Neumann series [cf. Eq. (25)] and the low-energy expansion [Eq. (29)] are in excellent agreement with the numerical results in their respective regimes of validity.

The consideration of this Section can be straightforwardly generalized to account for multiple pair-breaking modes: to determine the quasiparticle density $x_{qp}$, it suffices to replace

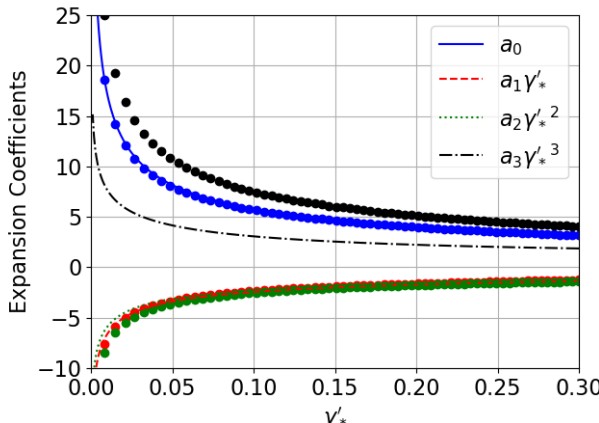

Figure 4: Coefficients of the low-energy expansion for $\phi$, Eq. (29). For better visibility, the products $a_l(\gamma'_*)^l$, $l = 0$ to 3, are plotted. Points are calculated using the numerical result for $\phi$ in Eq. (31), while lines using the approximate Neumann series as described in the text.

the right-hand side of Eq. (16) with its sum over all modes. Then, Eq. (19) [or equivalently Eq. (23)] being linear in the distribution function, it can be solved for the contribution $f_i(\xi_i \gamma_i)$ of each pair-breaking mode $i$ as for a single mode. The distribution function $f$ is finally given by the sum over all $f_i$.

## 3.1 Validity of assumptions

In obtaining the approximate solution for the distribution function, we made a number of assumptions whose validity depend on parameters such as the recombination rate $r$ and the quasiparticle-phonon coupling constant $c_{Phot,PB}^{QP}$; for aluminum, as order of magnitude we take $r \simeq 10^7$ Hz [11], and we estimate in Appendix A that for resonators of this material $c_{Phot,PB}^{QP} \simeq 10$ Hz. The condition in Eq. (18) can be interpreted as a bound on the number of pair-breaking photons, which with these parameters reads $\bar{n}_{PB} \ll 10^6 \xi/\Delta$. Even for pair-breaking photons near the threshold ($\xi \to 0$), this conditions is in practice always satisfied for $\bar{n}_{PB} \lesssim 1$: while for subgap photons high quality factors of order $10^6$-$10^7$ are possible [8, 18], the quality factor of above-gap modes is more than two orders of magnitude smaller, see Appendix A; this implies $\xi/\Delta > 10^{-5}$, since the frequency of the pair-breaking mode is known with relative precision given by the inverse quality factor. Alternatively, for pair-breaking photons of sufficient energy, $\xi/\Delta > 0.1$, the bound on the photon number becomes $\bar{n}_{PB} < 10^5$. We note that if Eq. (18) holds, then we also find from Eq. (16) that $x_{qp} \ll 1$; this ensures that the suppression of the gap $\delta\Delta/\Delta \simeq x_{qp}$ can also be neglected for the purposes of this section.

We now turn to the validity of Eq. (13), that is, negligibility of photon-mediated recombination. Setting $\varepsilon = \omega_{PB} - \Delta - E$, in the limit $\varepsilon \to 0$ (i.e., $\epsilon \to \xi$) we can rewrite that inequality in the form

$$\frac{\varepsilon}{\xi} \gg \left(\frac{a_0}{2\pi}\right)^2 \left[\frac{c_{Phot,PB}^{QP}(1 + \bar{n}_{PB})}{r}\right]^2, \tag{32}$$

where we used Eqs. (21) and (29). For $\bar{n}_{PB} \gg 1$ and assuming $\gamma'_* \gtrsim 1$, we have $a_0 \simeq 1$ and, for the parameter discussed above we have $\varepsilon/\xi \gg 10^{-14}\bar{n}_{PB}^2$; even at the upper bound $\bar{n}_{PB} \simeq 10^5$ determined in the previous paragraph, the right-had side is small, of order $10^{-4}$.

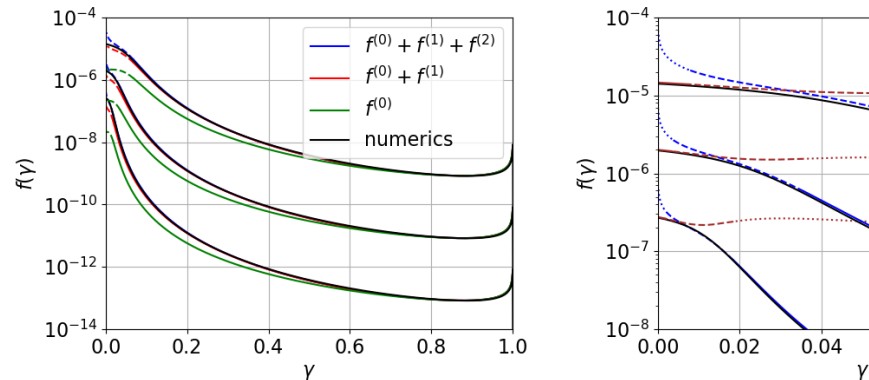

Figure 5: Shape of the quasiparticle distribution for different pair-breaking photon numbers and $\omega_{PB} = 2.8\Delta$. Numerical results are displayed in black, different orders of the Neumann series [cf. Eq. (25)] in green ($j = 0$), red ($j \leq 1$) and blue ($j \leq 2$). The different photon numbers correspond to (top to bottom) $c_{Phot,PB}^{QP} \bar{n}_{PB} = 10^{-2}, 10^{-4}$, and $10^{-6}$ Hz; other parameters are $\tau_0 = 63$ ns, $\Delta = 189\,\mu$eV, and $\tau_l = 0$, resulting in effective generation temperatures $\bar{T}_B \simeq 180$ mK, 150 mK, and 130 mK, respectively [c.f. Eq. (34)]. The bottom plot zooms into the low-energy region. Here, the Neumann series (up to second order) has been displayed with blue solid lines for $\gamma > \gamma_*'$, blue dashed lines for $\gamma_*' > \gamma > \gamma_*$ and blue dotted lines for $\gamma < \gamma_*$, while the low energy expansion [Eq. (29)] with solid brown lines for $\gamma < \gamma_c$, dashed lines for $\gamma_c < \gamma < \gamma_*'$, and dotted lines for $\gamma > \gamma_*'$.

In the opposite regime $\bar{n}_{PB}, \gamma_*' < 1$, the first factor on the right-hand side is of order unity (as can be verified by inspection of Fig. 4 or by using the approximate analytical expression for $a_0$ given previously) and the second factor is $\sim 10^{-12}$, meaning that the requirement becomes significantly less stringent. Therefore, as mentioned at the end of Sec. 2, recombination by photon emission can be ignored.

At the beginning of this section, we assumed that there are no other modes beside that of the pair-breaking photons by setting $c_{Phot}^{QP} = 0$. If such a mode is present ($c_{Phot}^{QP} > 0$), even if it is unpopulated ($\bar{n} = 0$) it can in principle affect the distribution function, since quasiparticles could relax by emitting a photon of energy $\omega_0$. However, similarly to the process of photon-mediated recombination considered in the previous paragraph, we now show that we can ignore this relaxation mechanism. Indeed, this mechanism is accounted for by the term proportional to $\bar{n} + 1$ in the second line of Eq. (3), and the corresponding relaxation rate diverges for $\epsilon \to \omega_0$. We compare this rate to that of relaxation by phonon emission, Eq. (9) (assuming that $1/\tau_{e,n}^{qp}(\omega_0) > 1/\tau_r^{qp}$, as we expect to be the case at low temperature and hence quasiparticle density); we find the condition

$$c_{Phot}^{QP} \sqrt{\frac{2\Delta}{\tilde{\varepsilon}}} \ll \frac{128}{105\sqrt{2}} \left(\frac{\Delta}{T_c}\right)^3 \left(\frac{\omega_0}{\Delta}\right)^{7/2} \frac{1}{\tau_0}, \tag{33}$$

where $\tilde{\varepsilon} = \omega_0 + \Delta - E \ll \omega_0$. This inequality is equivalent to $\tilde{\varepsilon}/\Delta \gg (\Delta/\omega_0)^7 (105 c_{Phot}^{QP}/16 r_0)^2$, and using $c_{Phot}^{QP} \simeq 1$ Hz (see Appendix A), we estimate that the second factor on the right-hand side is small, of order $10^{-13}$; while the first factor could in principle be large, since $\omega_0 \ll 2\Delta$, in practice the resonators are designed to have modes in the frequency range above at least a couple of GHz, in which case $(\Delta/\omega_0)^7 \lesssim 10^9$. Hence we conclude that the above condition is violated only for energy so close to $\Delta + \omega_0$ as to not affect the quasiparticle distribution. In fact, the numerical solutions displayed in Fig. 5 were obtained with $c_{Phot}^{QP} = 1$ Hz

and $\omega_0/2\pi = 4.84\,\text{GHz}$ and show no significant deviation from the analytical approximation derived assuming $c_{Phot}^{QP} = 0$.

Finally, we also assumed that we can ignore phonons by setting $T_B = 0$ and $\tau_l = 0$. Even within the assumption of fast thermalization ($\tau_l \ll \tau_0^{PB}$), thermal phonons with $T_B > 0$ can generate quasiparticles by breaking Cooper pairs; therefore, a necessary condition to ignore them is that this generation mechanism gives a negligible contribution to the quasiparticle density. By comparing the second line in Eq. (16) (that is, the quasiparticle density generation rate due to pair-breaking photons) to the thermal phonon generation rate [see text after Eq. (10)], we define the crossover (or effective generation) temperature

$$\bar{T}_B = \frac{2\Delta}{W\left(\frac{4r_0}{c_{Phot,PB}^{QP}\bar{n}_{PB}}\frac{\Delta}{\xi}\right)}, \tag{34}$$

with $W$ with the Lambert (product logarithm) function. For $T_B > \bar{T}_B$ thermal phonons are the main source of quasiparticles, while for $T_B < \bar{T}_B$, the dominant generation mechanism is photon pair breaking and $\bar{T}_B$ gives the temperature at which quasiparticles in thermal equilibirum would have the same density as those generated by the photons. For the typical parameters discussed above ($r_0 = 10^7$ Hz, $c_{Phot,PB}^{QP} = 10\,\text{Hz}$), even for photons with frequencies close to the pair-breaking threshold, $\xi/\Delta = 10^{-3}$, and a low occupation probability, $\bar{n}_{PB} = 10^{-10}$, we estimate a relatively high crossover temperature $\bar{T}_B > 100\,\text{mK}$. In fact, for the crossover temperature to go below e.g. 20 mK the photon occupation probability should be extremely small, $\bar{n}_{PB} < 10^{-84}$, so ignoring generation by thermal phonons should likely be a good approximation for typical low-temperature experiments; in our examples in the plots in Fig. 5 we have assumed $\bar{n}_{PB} \geq 10^{-7}$.

While the condition $T_B < \bar{T}_B$ implies that the density is not affected by thermal phonons, it is not sufficient to ensure that they do not alter the shape of the quasiparticle distribution. A more stringent condition is found by requiring that the rate of phonon absorption at the gap is small compared to the recombination rate [as defined in Ref. [17], the former rate is approximately given by $3/\tau_{e,n}^{qp}(T_B)$], or equivalently by requiring that the typical energy gained by absorbing a phonon is small compared to the width of the distribution in the absence of phonons, $T_B < \gamma_*'\xi$. These requirements can be expressed as $x_{qp} > (T_B/\Delta)^{7/2}$. With the same parameters used above and assuming $\xi > 0.1\Delta$, for $T_B = 10\,\text{mK}$ we estimate, using Eq. (16), that the condition is satisfied for $\bar{n}_{PB} > 10^{-8}$. The bound becomes more stringent as the phonon temperature increases; we do not explore this higher temperature regime further here, as we focus next on a competing mechanism affecting the distribution shape, namely the absorption and emission of non-pair-breaking photons.

The results of this Section show that at low temperature a small number or photons above the pair-breaking threshold can lead to experimentally relevant quasiparticle densities. In fact, using Eqs. (16) and (21) one can show that if both $\xi$ and $\gamma_*'$ are not too small compared to unity, then $f(0)$ (cf. Fig 5) is of the same order of magnitude as the normalized density $x_{qp}$, which in various experiments has been estimated to range between $10^{-9}$ and $10^{-5}$ (see [31] and references therein). Even the width $\sim \gamma_*'\xi$ of the quasiparticle distribution can be determined by the interplay between pair breaking by photons and recombination, rather than by absorption of thermal phonons. This could happen in particular in superconducting qubits, for which pair-breaking photons can contribute directly to qubit transitions as well as to quasiparticle generation [32, 33], although the actual shape of the distribution function is likely affected by the fact that typically the films forming the qubit's junction have different gaps. While qubits are by necessity operated in a regime where at most a few non-pair-breaking photons are present, this is not the case for resonators, where the large number of non-pair breaking photons can determined the distribution's shape, as we discuss next.

# 4 Non-pair-breaking photons

When both modes above and below the pair-breaking threshold are populated, $\bar{n}, \bar{n}_{PB} > 0$, various regimes are possible. For instance, for $\bar{n}$ sufficiently small compared to $\bar{n}_{PB}$ we can expect the quasiparticle distribution function to be mainly determined by the pair-breaking photons; then the effect of the non-pair-breaking ones can be treated perturbatively. Still, we can expect the effect to be different depending on the frequency $\omega_0 < 2\Delta$ satisfying $\omega_0 < \gamma_c \xi$, $\gamma_c \xi < \omega_0 < \xi$, or $\omega_0 > \xi$ (we have assumed $\gamma_c < 1$, otherwise $\omega_0$ should be compared to $\xi$ only). In the opposite regime of sufficiently large $\bar{n}$, the distribution function dependence on energy is instead mainly due to the below-threshold photons and the pair-breaking one mostly contribute to the overall quasiparticle density. In what follow, we examine in more detail some (but not all) of these many regimes.[1]

## 4.1 Perturbative regime

We begin by investigating the effect of a small number of non-pair-breaking photons of frequency $\omega_0 < 2\Delta$ on the quasiparticle distribution function. We assume that the approximations employed in Sec. 3 are still applicable, so that the steady-state kinetic equation becomes [cf. Eq. (19)]

$$0 = St^{Phon}\{f, n\} + St^{Phot}_{PB,g}\{f, \bar{n}_{PB}\} + St^{Phot}\{f, \bar{n}\}, \tag{35}$$

with $St^{Phot}\{f, \bar{n}\}$ given by Eq. (3). Simplifying that equation in the regime of low quasiparticle energies ($\epsilon < \Delta$) and densities ($f \ll 1$) we get

$$St^{Phot}\{f, \bar{n}\} \simeq c^{QP}_{Phot} \sqrt{\frac{2\Delta}{\epsilon + \omega_0}} [f(\epsilon + \omega_0)(\bar{n} + 1) - f(\epsilon)\bar{n}]$$
$$+ c^{QP}_{Phot} \sqrt{\frac{2\Delta}{\epsilon - \omega_0}} [f(\epsilon - \omega_0)\bar{n} - f(\epsilon)(\bar{n} + 1)], \tag{36}$$

where the last line vanishes for $\epsilon < \omega_0$.

In the previous section we have considered already when the effect of this term is negligible in the case $\bar{n} = 0$ by focusing on the decay rate associated with the last term on the right-hand side of Eq. (36) [a finite $\bar{n}$ adds the factor $(1 + \bar{n})$ to the left-hand side of condition (33)]. For finite but small occupation, $\bar{n} \lesssim 1$, to find a necessary condition enabling us to treat the non-pair-breaking photons as a perturbation we consider the absorption rate arising from the second term in the first square brackets, which is highest at the lowest energies $\epsilon < \omega_0$ at which the second line of Eq. (36) is zero [at higher energies, the photon absorption accounted for by the first term in the second square brackets should also be considered]; we then compare that rate to the recombination rate, Eq. (10), to find the condition

$$\bar{n} \ll \frac{r x_{qp}}{c^{QP}_{Phot}} \sqrt{\frac{\omega_0}{2\Delta}}, \tag{37}$$

where $x_{qp}$ depends on $\bar{n}_{PB}$ as follows from Eq. (16). Assuming the inequality holds, we write the quasiparticle distribution in the form

$$f(\epsilon) = f_0(\epsilon) + \delta f(\epsilon), \tag{38}$$

---

[1] Previous works [49,50] considered the population of both modes in order to evaluate the absorption efficiency of the pair-breaking photons for detector applications. The kinetic equations are solved there numerically assuming a broadened density of states and using a discretization procedure [51] that, to our understanding [17], violates quasiparticle number conservation; therefore, we do not attempt to compare the results in those articles to ours.

where $f_0$ is the steady-state distribution in the absence of low-energy photons, $\bar{n} = 0$, and the kinetic equation can be rewritten as

$$St^{Phot}\{f, \bar{n}\} + St^{Phon}\{\delta f\} = 0, \qquad (39)$$

where in the phonon collision integral we dropped the explicit dependence on the phonon distribution as we assume zero temperature, $n = 0$ (this implies in particular $St_g^{Phon} = 0$). Although the photon collision integral in Eq. (3) accounts only for emission or absorption of a single photon at a time, consecutive absorption processes uninterrupted by phonon emission lead to peaks in the energy distribution function at multiples of the photon energy. Thus, we seek an expression for $\delta f$ in form of an expansion $\delta f = f_1 + f_2 + ...$ using an iterative approach to find $f_m$ once $f_{m-1}$ is known:

$$St^{Phot}\{f_{m-1}, \bar{n}\} + St^{Phon}\{f_m\} = 0, \qquad (40)$$

with $m = 1, 2, ....$

To find an explicit expression for $f_m$, $m \geq 1$, we ignore the in-scattering part of $St^{Phon}\{\delta f\}$ – that is, the first term on the right-hand side of Eq. (6); we will comment below on this step. We also assume $1/\tau_{e,n}^{qp}(\omega_0) > 1/\tau_r^{qp}$, so that for $\epsilon > \omega_0$ we can ignore the term proportional to $x_{qp}$ in $St^{Phon}$. Furthermore, we note that in $St^{Phot}$ there is at each iteration a term that diverge as $\epsilon \to m\omega_0$ originating from the last line in Eq. (36). Of the two terms in square brackets there, we expect the first one to be dominant; in other words, we assume

$$f_m(\epsilon) \gg f_m(\omega_0 + \epsilon)(\bar{n} + 1)/\bar{n}. \qquad (41)$$

With these simplifications, we find peaks for $\epsilon > m\omega_0$ with approximate shape

$$f_m(\epsilon) \simeq \left(\frac{T_*}{\omega_0}\right)^{6m} \sqrt{\frac{\omega_0}{\epsilon - m\omega_0}} f_0(\epsilon - m\omega_0)\left(\frac{\omega_0}{\epsilon}\right)^{7/2} \Pi_{j=1}^{m-1}\left(\frac{\omega_0}{\epsilon - j\omega_0}\right)^4, \qquad (42)$$

where [16, 17]

$$T_* = \left(\frac{105}{64} c_{Phot}^{QP} \bar{n}\tau_0 T_c^3 \Delta\omega_0^2\right)^{1/6}. \qquad (43)$$

The value of $T_*$ being larger than $\omega_0$ means that the non-pair-breaking photons are effective at heating the quasiparticles [16, 17]; similarly, here it can counteract the fast suppression of the amplitude of the peaks caused by the last two factors in Eq. (42) being approximately $1/m^{7/2}[(m-1)!]^4$ for $\epsilon \simeq m\omega_0$.

The result for the peaks in Eq. (42) is compatible with Eq. (41) so long as $\bar{n} > 1/(m+1)^4$. We can also estimate if a given peak is visible by comparing its amplitude (which we estimate at $\epsilon = m\omega_0 + \gamma'_*\xi$) to the value of $f_0$ at the peak's position. We thus find the "visibility condition"

$$\left(\frac{T_*}{\omega_0}\right)^{6m} \frac{1}{[(m-1)!]^4} \gg \frac{105}{16}\sqrt{\frac{2\gamma'_*\xi}{\Delta}}\left(\frac{\Delta}{\omega_0}\right)^4 \frac{x_{qp}}{a_0}, \qquad (44)$$

where we assumed $\omega_0 > \gamma'_*\xi$. Using this expression, one can for instance check under which conditions the first peak is visible while at the same time Eq. (37) holds. More interestingly, for a given choice of parameters as $m$ increases the left-hand side decreases, so there is a last visible peak which we denote with index $m_l$ (we have implicitly assumed that $m_l < \xi/\omega_0$, so that it is consistent to ignore occupation at energies $\epsilon > \xi$ as in Sec. 3). For $m > m_l$, it is reasonable to set $f_m = 0$; in turn, this makes it possible to understand why Eq. (42) gives a good description of the last peak but not necessarily of lower-index peaks. Indeed, in our derivation we have ignored the first term on the right-hand side of Eq. (6); considering the

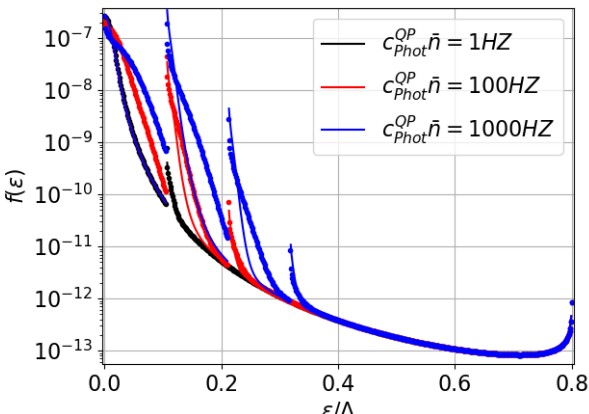

Figure 6: Distribution function in the presence of non-pair-breaking photons in the perturbative regime. The dots are obtained by solving the kinetic equation numerically with parameters $\omega_{PB} = 2.8\Delta$, $\tau_l = 0$, $\tau_0 = 63\,\text{ns}$, $\Delta = 189\,\mu\text{eV}$, $c_{Phot,PB}^{QP} \bar{n}_{PB} = 10^{-6}\text{Hz}$, and the values of $c_{Phot}^{QP}\bar{n}$ given in the inset. The solid lines are calculated using Eq. (42).

contribution of that term to Eq. (39) for $\epsilon > m\omega_0$, one can check that the integral between $\epsilon$ and $(m+1)\omega_0$ can be ignored compared to the first term in square brackets in Eq. (6). At energies just above $(m+1)\omega_0$, the integral is dominated by the $f_{m+1}$ term, which we ignore for $m = m_l$. On the other hand, taking for example $m = m_l - 1$, the $m_l$ peak gives a non-negligible contribution to the integral, a contribution that accounts for quasiparticle relaxing to lower energies by emitting phonons; this mechanism then leads to the lower-index peaks to be broader than what Eq. (42) predicts. In Fig. 6 we show with points the results of numerical solutions of the kinetic equation simulation for different number of non-pair-breaking photons. Only for the smallest number Eq. (37) holds; nonetheless, in all cases the number of visible peaks agrees with the expectation from Eq. (44) and the last visible peak is well described by Eq. (42), see the solid lines.

## 4.2 Heating regime

We next consider the regime in which there are many non-pair-breaking photons, such that they can heat the quasiparticles, $T_* > \omega_0$. Even if this condition is met, the pair-breaking photons could affect the shape of the distribution function; therefore, we additionally require that the generation of quasiparticles can be ignored at lowest order, so that the results of Refs. [16, 17] can be used as a starting point. Near energy $\xi$ we can approximate the photon generation collision integral, Eq. (4), as

$$St_{PB,g}^{Phot} \simeq c_{Phot,PB}^{QP} \bar{n}_{PB} \frac{\xi}{\sqrt{2\Delta(\xi - \epsilon)}}, \tag{45}$$

which diverges as $\epsilon \to \xi$; however, since the width of the peaks in the distribution function is limited by the photon frequency $\omega_0$, for our estimates we replace $\xi - \epsilon \to \omega_0$. If the shape is determined by the balance between phonon emission and the absorption/emission of non-pair-breaking photons, the above contribution from $St_{PB,g}^{Phot}$ should be small in comparison to, for instance, the phonon collision integral, or more precisely the term proportional to $(\epsilon/\Delta)^{7/2}$ in Eq. (6), where we can use [17]

$$f(\epsilon) \simeq \frac{3 \times 2^{1/6}}{4.2\sqrt{2\pi}} x_{qp} \sqrt{\frac{\Delta}{\epsilon}} e^{-(\epsilon/T^*)^3/3}, \tag{46}$$

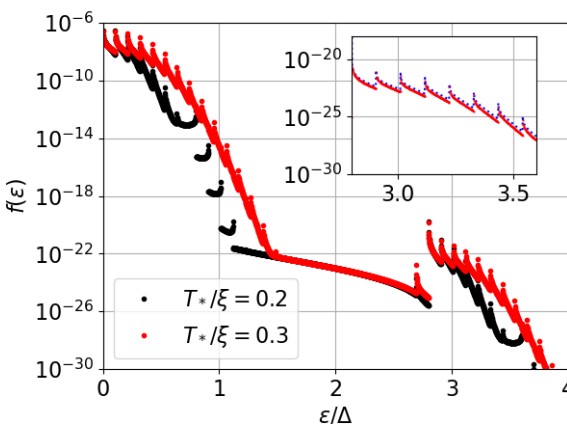

Figure 7: Distribution function in the crossover from perturbative to heating regimes as identified by Eq. (47). We use the same parameters as in Fig. 6 except for the photon number $\bar{n}$, given in the bottom left inset in terms of the ratio between $T_*$ of Eq. (43) and $\xi$ of Eq. (17). The top right inset shows the second peak at energies $\epsilon > \omega_{PB}$, with the blue dashed curve corresponding to Eq. (48) where $f(\epsilon)$ is the numerically calculated distribution function at low energies.

for $\epsilon = \xi$. Assuming that the quasiparticle density is determined by the photon pair-breaking mechanism and hence given by Eq. (16), we arrive at the condition

$$\frac{T_*}{\xi} \gtrsim \left\{ -3\ln\left[ 3.27 x_{qp} \left(\frac{\Delta}{\xi}\right)^3 \sqrt{\frac{\Delta}{\omega_0}} \right] \right\}^{-1/3}. \tag{47}$$

For the parameters of Fig. 6 this correspond to $T_*/\xi \gtrsim 0.29$, while for the highest photon number there we estimate $T_*/\xi \simeq 0.1$. Even doubling the value of $T_*$ so that $T_*/\xi = 0.2$ (or $c_{Phot}^{QP}\bar{n} \simeq 8.2 \times 10^4\,\text{Hz}$), see Fig. 7, the considerations of the previous subsection are still valid with minor modifications. For example, according to Eq. (44) the $m = 7$ peak should be the last visible one, but since that peak is close in energy to $\xi$, the right-hand side of that equation should be multiplied by a factor of order $\sqrt{\xi/\omega_0}$; including this factor, we find that the inequality holds only weakly. Further increasing $T_*$ above the threshold value by setting $T_*/\xi = 0.3$ ($c_{Phot}^{QP}\bar{n} \simeq 9.3 \times 10^5\,\text{Hz}$), the shape of the distribution function follows the predictions of Ref. [17] up to energies beyond $\xi$.

In Fig. 7 it is evident that the distribution function changes qualitatively at a crossover energy $\epsilon_c > \xi$, becoming mostly smooth in the range $\epsilon_c \lesssim \epsilon \lesssim \omega_{PB}$. We ascribe this smooth part to "excess" quasiparticles relaxing by phonon emission (and/or scattering with non-pair-breaking photons), the excess being understood as the second broad peak at energies above $\omega_{PB}$. As mentioned in Sec. 3, this second peak is due to the absorption of pair-breaking photons, and can be found by following the approach of Ref. [16] [cf. Eq. (8) there]:

$$f(\omega_{PB} + \epsilon) \simeq \frac{\bar{n}_{PB} f(\epsilon)/(1 + \bar{n}_{PB})}{1 + \frac{1}{12} \frac{r_0}{c_{Phot,PB}^{QP}} \left(\frac{\omega_{PB}}{\Delta}\right)^3 \sqrt{\frac{2\epsilon}{\Delta} \frac{\Delta + \omega_{PB}}{2\Delta + \omega_{PB}}}}. \tag{48}$$

We compare this equation to numerical calculations in the top right inset of Fig. 7. These modifications affect the distribution only at the relatively large energies $\epsilon > \epsilon_c$ and, as explained in Sec .2, do not contribute significantly to quasiparticle creation even when considering a finite phonon thermalization time (see also the next subsection). Furthermore, since usually $r_0 \gg c_{Phot,PB}^{QP}$, the second peak is much smaller than the first one and thus does not appreciably affect measurable properties like the quality factor of a resonator.

## 4.3 Quasiparticle density and its fluctuations

So far we have assumed that quasiparticle generation by phonons can be ignored. Once we enter the heating regime, however, the distribution function is not significantly affected by the pair-breaking photons, so that the latter can only influence the overall quasiparticle density. Therefore, in this regime we can extend the generalized Rothwarf-Taylor equation determining the quasiparticle density by including contributions from both pair-breaking phonons and photons,

$$\frac{dN_{\text{qp}}}{dt} = G_{T_B} + G_{\bar{T}_B} + G(T_*/\Delta)N_{\text{qp}} - RN_{\text{qp}}^2, \tag{49}$$

with $G_{T_B}$ of Eq. (11) and $G_{\bar{T}_B}$, accounting for pair-breaking photons, obtained from $G_{T_B}$ by replacing $r \to r_0$ and $T_B \to \bar{T}_B$, see Eq. (34); this is equivalent to $G_{\bar{T}_B} = 2\pi\rho_F c_{Phot,PB}^{QP} \bar{n}_{PB}\xi$. The term linear in $N_{\text{qp}}$ depends on $T_*$ because it describes generation by nonequilibrium phonons emitted by quasiparticles heated by the non-pair-breaking photons to energies above $3\Delta$ [see diagram h) in Fig. 2 and Ref. [17]]:

$$G(x) = 0.21r\frac{\tau_l}{\tau_0^{PB}}x^{9/2}e^{-\sqrt{14/5}\,x^{-3}}. \tag{50}$$

Equation (49) can be derived by multiplying Eq. (1) by $\rho(E)$ and integrating it over energy; in the collision integrals, the (nonequilibrium) phonon distribution can be expressed in terms of the quasiparticle distribution, while the energy dependence of the latter, but not its normalization, are determined by the balance between phonon scattering processes and non-pair-breaking photon absorption; a detailed derivation is given in Ref. [17].

Clearly, as far as the density is concerned, pair-breaking phonons and photons play the same role; for $r \sim r_0$, as discussed after Eq. (34) one of the two generation mechanisms is dominant, depending on which of the two temperatures, $T_B$ or $\bar{T}_B$, is the largest. Then defining $T_G = \max\{T_B, \bar{T}_B\}$ and $T_B^* = T_*^3/\Delta^2$ [17], we can distinguish two steady-state regimes, in analogy to those discussed in Ref [17] in the absence of pair-breaking photons. At high heating, $T_B^* > T_G$, we can neglect the first two terms on the right-hand side and find $N_{qp,0} \simeq G(T_*/\Delta)/R$, independent of phonon temperature and number of pair-breaking photons (we use subscript 0 to denote the steady state); for low heating, $T_B^* < T_G$, we can neglect the term linear in $N_{qp}$ in Eq. (49) and the density is approximately $N_{qp,0} \simeq \sqrt{(G_{T_B} + G_{\bar{T}_B})/R}$, independent of the number of non-pair-breaking photons $\bar{n}$. In both regimes, the dependence of the density on bath temperature are similar: at low temperature – $T_B < \bar{T}_B$ for low heating and $T_B < T_B^*$ for high heating – the density is approximately constant, while above these crossover temperatures it increases exponentially, being approximately the same as in thermal equilibrium. As an example, in Fig. 8 we plot the density as function of temperature for a few values of $\bar{T}_B$ in the low-heating regime; the behavior resembles that in the high-heating regime, see Ref. [17].

Interestingly, the two regimes display quantitatively different power spectral densities for quasiparticle number fluctuations. In general, as shown in Appendix C the spectral density (per unit volume) can be written in the form

$$S_N(\omega) = \frac{8RN_{qp,0}^2\bar{\tau}_r^2}{1 + (\omega\bar{\tau}_r)^2}, \tag{51}$$

where $\bar{\tau}_r = 1/[2RN_{qp,0} - G(T_*/\Delta)]$ is the relaxation time of quasiparticle number fluctuations and we assumed fast phonon dynamics, $\bar{\tau}_r \gg \tau_0^{PB}, \tau_l$. At low heating, we have $\bar{\tau}_r \simeq 1/2RN_{qp,0}$ and we recover the result of Ref. [34]. At high heating $\bar{\tau}_r \simeq 1/RN_{qp,0}$, which modifies the relation between measurable quantities such as $\bar{\tau}_r$ and $S_N(0)$ and quantities than can be derived from them, such as the steady-state quasiparticle density.

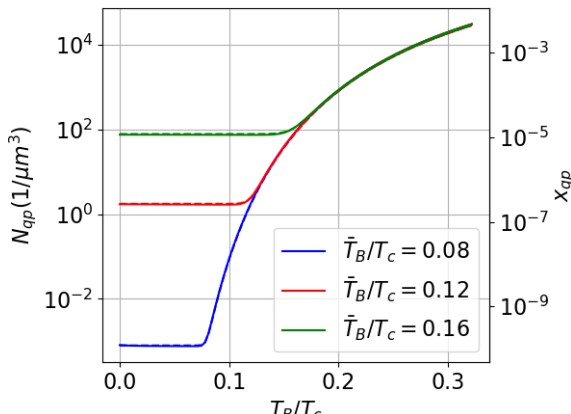

Figure 8: Quasiparticle density for fixed numbers of pair-breaking photons (see parameter in the inset) as function of temperature in the low-heating regime in which non-pair-breaking photons have negligible influence on the density. Results calculated from numerical solutions to the kinetic equation are displayed with full lines and the corresponding analytical ones obtained by solving Eq. (49) in the steady state with dashed lines.

It should be noted that in experiments the fluctuations measured are not directly those of the quasiparticle number, but those of phase and amplitude of the complex transmission. Therefore, comparison to experiments requires knowledge of the responsivities [9, 35]

$$\frac{dA}{dN_{qp}} = -\frac{2\alpha Q}{\sigma_2}\frac{d\sigma_1}{dN_{qp}} \simeq -\frac{2Q}{N_{qp}Q_{i,qp}}\,, \tag{52}$$

$$\frac{d\theta}{dN_{qp}} = \frac{4Q}{\omega_0}\frac{d\delta\omega_0}{dN_{qp}} \simeq 4Q\frac{\delta\omega_0}{N_{qp}\omega_0}\,, \tag{53}$$

where $Q$ is the loaded quality factor, $\alpha$ the kinetic inductance fraction, $\sigma_{1(2)}$ the real (imaginary) part of the ac conductivity, $Q_{i,qp}$ the contribution of quasiparticles to the internal quality factor, and $\delta\omega_0$ the frequency shift. For a coupling-limited quality factor, both responsivities are approximately independent of quasiparticle number. However, in the heating regime the responsivities depend on the photon number $\bar{n}$ via $Q_{i,qp}$ and $\delta\omega_0$, as investigated in Ref. [17] – see also the next section.

In closing this section, we remark that the considerations presented in this work for the heating regime can be straightforwardly extended to the case of multiple pair-breaking modes (cf. Ref. [36]) by introducing more sources like the term $G_{\bar{T}_B}$ in Eq. (49).

## 5 Comparison to experiments

We now turn our focus to the implications of the results of the previous sections for experiments. We reconsider in particular the measurement of quality factor and resonant frequency as functions of temperature and for various readout powers reported in Ref. [18]. We reproduce in Fig. 9 the internal $Q$-factor data from that reference together with fit lines calculated using the theory of quasiparticle heating of Ref. [17], which does not include the effect of pair-breaking photons, with the addition of an "extrinsic" dissipation mechanism of unknown origin; this second mechanism is needed in order to capture the low temperature saturation

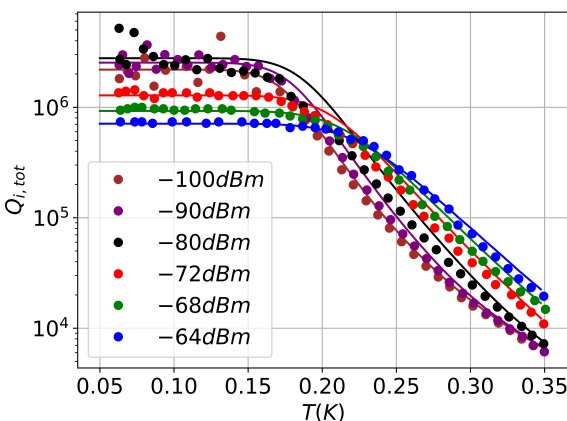

Figure 9: Quality factor vs temperature for different readout powers. The experimental points are taken from Ref. [18]. The solid lines are based on Eq. (54) and are calculated using the following parameters: $\Delta = 189\,\mu\text{eV}$, $c_{Phot}^{QP} = 58\,\text{mHz}$, $\tau_0 = 63\,\text{ns}$, $\omega_0/2\pi = 5.3\,\text{GHz}$, $\alpha = 0.13$, $\tau_l = 170\,\text{ps}$. More details can be found in Ref. [17].

of the $Q$-factor. Concretely, the inverse internal quality factor is assumed to be of the form

$$\frac{1}{Q_{i,\text{tot}}} = \frac{1}{Q_{i,\text{ext}}} + \frac{1}{Q_{i,qp}}, \tag{54}$$

where, at leading order in $T_*/\Delta$, we have [17]

$$Q_{i,qp} = \frac{\sigma_2}{\alpha\sigma_1} \simeq \frac{4.1}{\alpha}\frac{2\rho_F\Delta}{N_{qp}}\frac{\Delta}{\omega_0}\left(\frac{T_*}{\Delta}\right)^{3/2}, \tag{55}$$

leading to a linear relation between inverse quality factor and quasiparticle density $N_{qp}$, with a slope that decreases with increasing readout power (the condition $T_* > \omega_0$ ensures that the quality factor is not affected by the detailed shape of the peaks on the scale $\omega_0$, but only on the overall width $T_0$ of the distribution function). In other words, since at low heating the quasiparticle density is independent of the photon number $\bar{n}$, the quality factor is expected to increase with readout power, as $\bar{n}$ and hence $T_*$ increase; indeed, when the coupling quality factor is small, $Q_c \ll Q_i$, we have $\bar{n} = 2Q_c P_{\text{read}}/\omega_0^2$ [17]. While the increase of $Q_i$ with power agrees with observations at sufficiently high temperatures, it is qualitatively inconsistent with measurements at the lowest temperatures (crossover to the high-heating regime with decreasing temperature cannot explain the data, see Ref. [17]). We emphasize that $Q_{i,\text{ext}}$ is introduced purely in a phenomenological way, and ranges from approximately $2.5 \times 10^6$ at low power to $0.7 \times 10^6$ at high power. The decrease at higher power excludes the standard explanation of the low-temperature value of $Q_{i,\text{ext}}$ as originating from two-level systems, since the saturation of the latter with increasing power would lead to an increase of the quality factor [19].

To try and explain the low-temperature behavior of the quality factor, we now consider the effect of pair-breaking photons, assuming we remain in the low-heating regime. The quasiparticle density can be strongly affected by a small number of such photons, see the discussion after Eq. (34). In fact, in a series of works [37,38] radiation emitted by a higher temperature (3-4 K) stage of the refrigerator was identified as a source of quasiparticle generation, and at least partially mitigated by using a 'box in a box' design. Furthermore, to reduce the influence of pair-breaking photons propagating through coaxial cables, filters have been used at the cold stage that, extrapolating measurements below $\sim 10\,\text{GHz}$ to twice the gap, are expected to

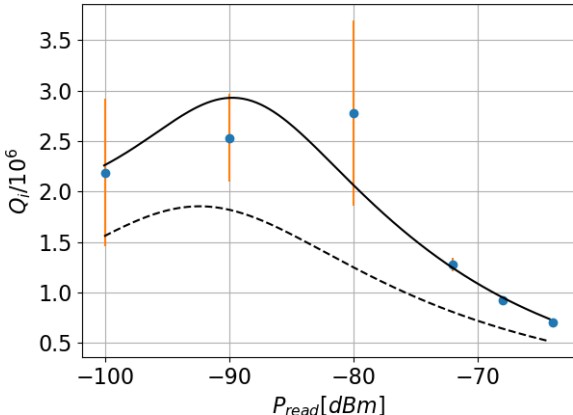

Figure 10: Low-temperature $Q$-factor vs readout power. Experimental data points are obtained by averaging the low temperature ($T_B < 150\,\text{mK}$) data in Ref. [18] and each error bar shows the corresponding standard deviation; the value of the data points coincide with those of $Q_{i,\text{ext}}$ [cf. Eq. (54)] used in Fig. 9. The solid line is obtained from numerical solutions to the kinetic equation ignoring thermal phonons (that is, setting $T_B = 0$) and using parameters $\omega_{PB} = 2.8\Delta$, $c_{Phot,PB}^{QP} = 27 c_{Phot}^{QP}$, $\tau_0 = 20\,\text{ns}$, $\bar{n}_{PB,0} = 2 \times 10^{-4}$, and $\mu = 10^{-9}$ [see Eq. (56)]; other parameters as in Fig. 9. The dashed line shows the analytical estimate, Eq. (55).

suppress the number of pair-breaking photons by almost 4 orders of magnitude. Despite such careful engineering, the "leakage" of pair-breaking photons from higher temperature stages is likely not completely eliminated. However, at low temperatures such leakage would result in a density of quasiparticles independent of $\bar{n}$ and of $T_B$ (up to $\bar{T}_B$, see Fig. 8), and the quality factor should still increase with $\bar{n}$, in contrast with observation.

Since the assumption of a fixed number of pair-breaking photons is inconsistent with experiment, we are lead to amend it by setting

$$\bar{n}_{PB} = \bar{n}_{PB,0} + \mu\bar{n}\,, \tag{56}$$

where the first term on the right-hand side accounts for the leakage from higher temperature stages discussed above. The second term introduces a dependence of $\bar{n}_{PB}$ on $\bar{n}$ and can originate from high-frequency noise produced by the source used to generate the signal probing the resonator. This dependence in turns implies that when $\bar{n} \gtrsim \bar{n}_{PB,0}/\mu$ the quasiparticle density increases with readout power even in the low-heating regime, because the second term on the right-hand side of Eq. (49) increases with $\bar{n}$. As shown in Fig. 10, with this assumption we can obtain a satisfactory fit to the measurements of Ref. [18] using reasonable parameters: the "leaked" photon number $n_{PB,0}$ is about one order of magnitude bigger than what one would expect assuming a higher-stage temperature of 3.5 K with a $10^{-4}$ attenuation factor for photons of energy $\omega_{PB} = 2.8\Delta$, but the discrepancy would drop to just a factor of 3 for photons at the pair-breaking threshold.[2] As for the value of $\mu = 10^{-9}$, it can be explained by a combination of $10^{-4}$ attenuation times a $10^{-5}$ ratio between power of the generated signal and of the high-frequency noise.[3]

---

[2]We note that $n_{PB,0}$ could also originate from even higher-temperature stages with higher attenuation, so the independence of $e.g.$ $\bar{\tau}_r$ on the temperature of the few Kelvin stage [9] does not exclude such leakage.

[3]Spectral purity of signal generators at few GHz frequencies can range between -30 dBc to -140 dBc, depending on whether harmonics, nonharmonics, or wideband noise are considered, indicating that power ratios up to $10^{-3}$ cannot be excluded.

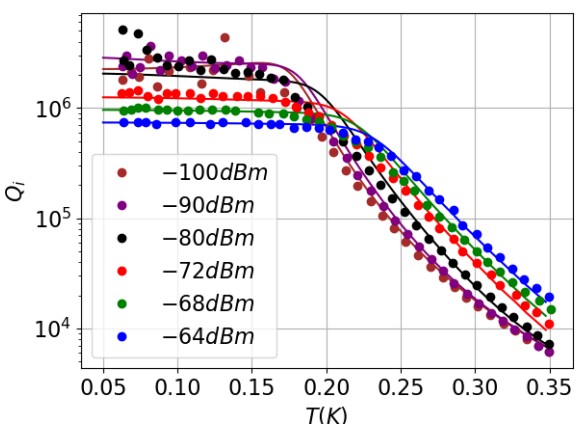

Figure 11: Quality factor vs temperature for different readout powers. As in Fig. 9, the experimental points are from Ref. [18]. Here the solid lines include only the quasiparticle contribution, with no extrinsic loss mechanisms. The lines are calculated using the same parameters as in Fig. 10 but, in contrast to Fig. 9, here we take into account effect of a power-dependent number of pair-breaking photons, see Eq. (56) and the discussion that follows it.

The assumption in Eq. (56) is independent of temperature, so we can employ it also when evaluating the temperature dependence of the quality factor, as shown in Fig. 11. We stress that while in Fig. 9 the value of $Q_{i,\text{ext}}$ is chosen independently for each of the six readout powers as to best fit the data, in Fig. 11 just the two parameters appearing in the right-hand side of Eq. (56) are used to fit all the data at the same time.[4] The agreement between theory and data in Figs. 10 and 11 thus support our proposed explanation for the physical mechanism responsible for the saturation of the quality factor in terms of excess pair-breaking photons.

We also remark that the way pair-breaking photons affect the quality factor is not equivalent to adding an extrinsic mechanism: the pair-breaking photons influence $Q_{i,qp}$ via the quasiparticle density $N_{qp}$, and in the low-heating regime we can write the latter in the form

$$N_{qp} = \sqrt{N_{T_B}^2 + N_{\bar{T}_B}^2}\,, \tag{57}$$

with $N_{T_B} = \sqrt{G_{T_B}/R}$ and the analogous definition for $N_{\bar{T}_B}$. Defining $Q_{i,T_B(\bar{T}_B)}$ by the replacement $N_{qp} \to N_{T_B}(N_{\bar{T}_B})$ in Eq. (55), we find that

$$\frac{1}{Q_{i,qp}} = \sqrt{\frac{1}{Q_{i,T_B}^2} + \frac{1}{Q_{i,\bar{T}_B}^2}}\,, \tag{58}$$

which should be contrasted with Eq. (54). Finally, we point out that the above analysis of the quality factor is not restricted to a single pair-breaking mode, but is valid in the heating regime if multiple pair-breaking modes (or even a broadband source) are present: the heating regime is by definition that in which the shape but not necessarily the normalization of the distribution function is determined by the interplay between non-pair-breaking photons and phonons, see Sec. 4.2; in this regime, the pair-breaking photons influence the quality factor only by contributing to the total quasiparticle density as in Eq. (57).

---

[4]Strictly speaking, there are two additional parameters, $c_{Phot,PB}^{QP}$ and $\xi$; however, only the product $c_{Phot,PB}^{QP} \bar{n}_{PB} \xi$ is relevant, as it enters $G_{\bar{T}_B}$, see text following Eq. (49). To arrive at order-of-magnitude estimates for the parameters in Eq. (56), we choose $c_{Phot,PB}^{QP}$ based on the considerations in Appendix A and the value of $\xi$ based on Ref. [21].

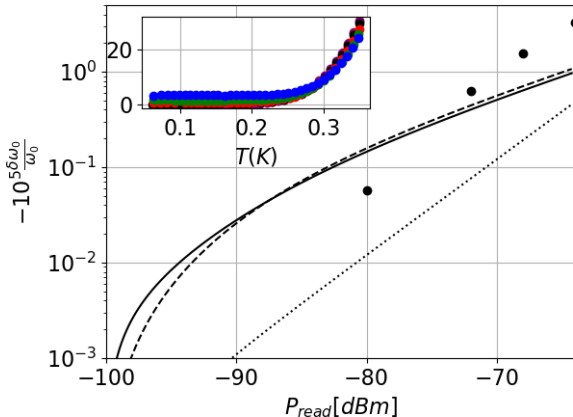

Figure 12: Main panel: Dots: relative shift in resonance frequency from Ref. [18] at low temperature, $T_B < 150\,\mathrm{mK}$, relative to the lowest readout power, $P_\mathrm{read} = -100\,\mathrm{dBm}$. Solid line: numerical calculation of the shift for $T_B = 0$ and other parameters as in Fig. 10. Dashed line: order-of-magnitude estimate from Eq. (59); both lines calculated as deviations from the value for $P_\mathrm{read} = -100\,\mathrm{dBm}$. Dotted line: prediction from Ref. [39] (corrected for the finite kinetic inductance fraction, see text). Inset: temperature evolution of the shift in linear scale (the colors are the same as in Fig. 11).

## 5.1 Frequency shift

The approach presented above to calculate the internal quality factor can also be applied to evaluate the shift in resonant frequency as function of the readout power. For thin-film resonators, the relative change in frequency is proportional to the kinetic inductance fraction times the relative change in the imaginary part of the ac conductivity [40]. For the latter quantity, we use here the order-of-magnitude estimate discussed in Ref. [17] to find for the relative shift

$$\left| \frac{\delta\omega_0}{\omega_0} \right| \simeq -\frac{\alpha}{2} x_{qP} \left\{ \left[ 1 - 0.42\frac{T_*}{\Delta} + 0.22\left(\frac{T_*}{\Delta}\right)^2 \right] + \frac{1}{2.1}\sqrt{\frac{2\Delta}{T_*}} \right\}. \tag{59}$$

The terms in square brackets originate from the suppression of the order parameter by the nonequilibrium quasiparticles, while the last term from the direct effect of the nonequilibrium distribution on the imaginary part of the ac conductivity. The latter term is in fact only a rough estimate, since for its actual calculation knowledge of the shape of the first peak (between energies $\Delta$ and $\Delta + \omega_0$) in the distribution function is needed [17].

Equation (59) predicts at low temperature a nonmonotonic dependence of the magnitude of the shift on readout power: at low power, $x_{qP}$ is independent of power as the quariparticle generation is dominated by the leakage term $n_{PB,0}$ in Eq. (56), while the factor in curly brackets decreases slowly with power. As power increases and the last term in Eq. (56) becomes relevant, the magnitude of the shift increases with power as the increase in $x_{qP}$ dominates on the decrease of the curly-bracket factor. In practice, however, one can measure the shift from the lowest readout power, not from zero power, so depending on the parameters the non-monotonic behavior might not be measurable. In fact, for the parameters used to fit the quality factor data the difference in relative shift between the two lowest powers is about $10^{-6}$ which, given quality factors of order $10^6$, is at the limit of being measurable. For higher powers, the magnitude of the shift increases, in qualitative agreement with the experiment, see Fig. 12. In that figure, we plot both the analytical estimate above and the numerically calculated relative

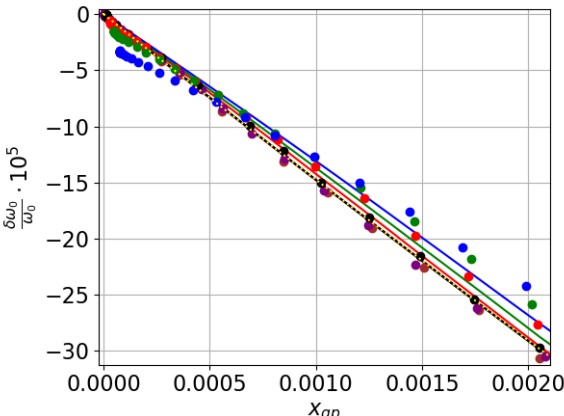

Figure 13: Dots: Experimental shift of resonance frequency from Ref. [18] (see inset in Fig. 12) vs quasiparticle density; the latter is obtained from the numerical solution to the kinetic equation with the same parameters as in the fits in Figs. 10 and 11. Solid lines: numerical results for the frequency shift calculated using the same parameters as for the density; readout powers increases from bottom to top (colors as in Fig. 11). Yellow dotted line: frequency shift assuming a thermal equilibrium distribution; for reference, $x_{qp} = 0.002$ corresponds to $T_B = 350\,\text{mK}$.

shift. Both curves deviate from the experimental data by less than a factor of 3.

We note that in Ref. [39] an alternative explanation for the shift was proposed: the authors argue that, in analogy with the case of a dc supercurrent, the non-pair-breaking photons cause a broadening of the peak in the superconducting density of states. This broadening effect can be characterized in terms of a depairing parameter $\tilde{\alpha}$ (not to be confused with the kinetic inductance fraction $\alpha$) which is proportional to the readout power and which causes a change in the kinetic inductance: $\delta L_k / L_k \simeq 4.85 \tilde{\alpha} / \Delta$ [39]. The relative shift is in turn proportional to the change in kinetic inductance, $\delta \omega_0 / \omega_0 = -\alpha \delta L_k / 2 L_k$; implicitly assuming $\alpha = 1$, good agreement was found between the theory of Ref. [39] and the experiment of Ref. [18]. However, the kinetic inductance fraction in the experiment can be estimated as $\alpha \sim 0.1$ [17], meaning that the theoretical result in fact underestimates the shift by approximately an order of magnitude compared to experiment. Interestingly, adding the contributions to the frequency shift of the two mechanisms improves agreement with the experimental data, but further investigation is needed to understand if in fact the two mechanisms coexist.

The inset in Fig. 12 displays the temperature dependence of the relative shift as extracted from the data of Ref. [18]. It is instructive to present the same data as function of normalized density $x_{qp}$ instead, since Eq. (59) predicts a linear relationship with a slope dependent on readout power. For low readout power, the experimental data and the numerical estimates are in reasonable quantitative agreement with each other and with the expectation for thermal equilibrium, see Fig. 13. As power increases, the qualitative trend of decreasing magnitude of the slope in the frequency vs density plot is evident in both data and numerics; although there is quantitatve discrepancy at the higher powers, we can conclude that a slope smaller in magnitude than the thermal equilibrium one is indicative of quasiparticle heating. Recently, such a reduced slope has been observed in a transmon qubit in which excess quasiparticles are generated due to the presence of an IR laser beam focused onto the qubit or the substrate next to it [41].

## 6 Conclusion and outlook

In this work, we extend previous analytical studies of the quasiparticle distribution in superconducting resonators in the presence of (sub-gap) photons and phonons [16, 17] to include the effect of a small number of photons of energy above the pair breaking threshold $2\Delta$. We derive expressions for the distribution's shape in the limit of low bath temperatures and low numbers of sub-gap photons, see Secs. 3 and 4.1, complementing the results obtained before in the opposite limit of large numbers of sub-gap photons (see also Sec. 4.2). In the latter regime, the quasiparticle density can be determined using a generalized Rothwarf-Taylor equation, Eq. (49); at typical temperatures of the phonon bath in the tens of millikelvin, we find that absorption of pair-breaking photons can be the dominant quasiparticle generation mechanism already at extremely low occupation $\bar{n}_{PB} \ll 1$. As a consequence, the saturation of the quality factor of superconducting resonators at low temperatures observed experimentally in Ref. [18] can be explained by assuming a low number of pair-breaking photons, if this number has two contributions: one originating from higher temperature stages of the cryogenic setup and the second from the microwave signal generator, see Eq. (56) and Figs. 10 and 11. In addition to contributing to losses, quasiparticles also cause a shift in the resonator frequency, analogously to the Kramers-Kronig relation between real and imaginary parts of response functions (cf. Ref. [42] for the case of qubits). Interestingly, plotting the frequency shift as function of the quasiparticle density can provide information on the distribution in energy of the quasiparticles, see Eq. (59) and Fig. 13.

Our results show that resonators can be used to probe the high-frequency (that is, above the gap) electromagnetic environment of superconducting circuits, in this respect complementing the use of qubits to probe low-frequency modes they are dispersively coupled to [43, 44]. The ability to investigate this environment can guide further efforts in reducing the detrimental impact of pair-breaking photons on the coherence time of superconducting qubits [21–23]. The change of resonator response with power and/or temperature is also used to explore non-quasiparticle loss mechanisms such as dielectric and two-level system losses [45–47], so our results can contribute to improving the characterization of superconducting materials.

## Acknowledgments

**Funding information** This work was supported in part by the German Federal Ministry of Education and Research (BMBF), funding program "Quantum Technologies – From Basic Research to Market," project QSolid (Grant No. 13N16149). G.C. acknowledges support by the U.S. Government under ARO grant W911NF2210257.

## A Coupling constant

In this Appendix, we estimate the coupling constant for the pair breaking photons $c_{Phot,PB}^{QP}$. The approach is the same used in Ref. [17] to find the coupling constant for the resonator's photons $c_{Phot}^{QP} \sim 1\,\text{Hz}$, giving the coupling constant in the form

$$c_{Phot,PB}^{QP}(\omega_{PB}) = \frac{\delta}{2Q'(\omega_{PB})}, \tag{A.1}$$

where

$$Q'(\omega_{PB}) = \frac{\sigma_2(\omega_{PB})}{\alpha(\omega_{PB})\sigma_N}, \tag{A.2}$$

is the quality factor with the real part $\sigma_1$ of the ac conductivity $\sigma = \sigma + i\sigma_2$ replaced by the normal-state conductivity $\sigma_N$ and $\alpha$ is the kinetic inductance fraction (that is, the ratio of kinetic to total inductance). For frequencies above $2\Delta$ and up to a few times $\Delta$, real and imaginary part of the conductivity are of comparable magnitude, leading to a frequency dependent $Q'$. Indeed, in the following we estimate the frequency dependence of the kinetic inductance fraction by the frequency dependence of the kinetic inductance of a thin film, *i.e.* assuming uniform current through the resonator cross section; as the width of the central strip is much larger than the penetration depth, we expect this to be a reasonable approximation. At small quasiparticle densities ($f \ll 1$), at leading order $\sigma_2$ is [48]

$$\frac{\sigma_2(\omega)}{\sigma_N} = \frac{1}{2}\left[\left(\frac{2\Delta}{\omega}+1\right)E(k') + \left(\frac{2\Delta}{\omega}-1\right)K(k')\right], \tag{A.3}$$

with $E$ and $K$ the complete elliptic integrals of the second and first kind, respectively, $k' = \sqrt{1-k^2}$, $k = |2\Delta - \omega|/|2\Delta + \omega|$; the left-hand side has the limiting forms $\pi\Delta/\omega$ for $\omega \ll 2\Delta$ and $\pi(\Delta/\omega)^2$ for $\omega \gg 2\Delta$.

The frequency dependence of the kinetic inductance can be obtained from the thin-film surface impedance $Z_s = 1/d\sigma(\omega)$ for a film of thickness $d$ small compared to the magnetic field penetration depth [40], giving the kinetic inductance per square as

$$L_k(\omega) = \frac{1}{\omega}\mathrm{Im}Z_s = \frac{\sigma_2(\omega)}{d\omega|\sigma(\omega)|^2}. \tag{A.4}$$

Assuming the kinetic inductance fraction to be small leads to $\alpha(\omega) \propto L_k$, which enables us to calculate the ratio between coupling strengths at different frequencies as

$$\frac{c_{Phot,PB}^{QP}}{c_{Phot}^{QP}} = \frac{|\sigma(\omega_0)|^2 \omega_0}{|\sigma(\omega_{PB})|^2 \omega_{PB}} \tag{A.5}$$

$$\simeq \frac{\pi^2\Delta^2}{\omega_{PB}\omega_0}\frac{\sigma_N^2}{|\sigma(\omega_{PB})|^2}, \tag{A.6}$$

where in the second line we have used that for $\omega_0 \ll 2\Delta$ we have $\sigma_1 \ll \sigma_2$ and the approximate value for $\sigma_2$ discussed after Eq. (A.3). In the limit $\omega_{PB} \gg 2\Delta$, the conductivity approaches its normal state value and hence $c_{Phot,PB}^{QP}/c_{Phot}^{QP} \simeq \pi^2\Delta^2/\omega_0\omega_{PB}$. Here we are interested into moderate frequencies, $\omega \lesssim 3.6\Delta$; in this range the kinetic inductance varies by less than a factor 2, and we can approximate the kinetic inductance fraction as frequency independent. Then we get $c_{Phot,PB}^{QP}/c_{Phot}^{QP} \sim \sigma_2(\omega_0)/\sigma_2(\omega_{PB}) \sim \omega_{PB}^2/\omega_0\Delta$. For a typical aluminium resonator with $\omega_0 \simeq 0.1\Delta$, this gives $c_{Phot,PB}^{QP} \sim (10^1 - 10^2)c_{Phot}^{QP}$ in this regime. Accordingly, for order-of-magnitude estimates we use a conservative value for the coupling constant for pair-breaking photons of order $c_{Phot,PB}^{QP} \sim 10\,\mathrm{Hz}$.

With the assumptions made in this Appendix, we also find that the ratio between the quality factors for above- and below-threshold frequencies is

$$\frac{Q(\omega_{PB})}{Q(\omega_0)} \simeq \frac{c_{Phot}^{QP}}{c_{Phot,PB}^{QP}}\frac{Q'(\omega_0)}{Q(\omega_0)}. \tag{A.7}$$

Since for our parameters the first factor on the right-hand side is of order $10^{-2} - 10^{-1}$ and the second factor is typically less than 1, this shows that the quality factor of pair-breaking modes is smaller than that of non-pair-breaking ones, as expected.

# B Neumann series

We give in this Appendix the $j = 1, 2$ terms in the Neumann series, Eq. (25), discussed in Sec. 3. For the $j = 1$ contribution, performing the integral in Eq. (26) using the simplified kernel in Eq. (27) we find

$$\phi^{(1)}(\gamma) = \frac{105}{128} \frac{5\gamma^3 - 6\gamma^2 - 8\gamma + 16}{8} \operatorname{arctanh}\left(\sqrt{1-\gamma}\right)$$
$$+ \frac{105}{128} \frac{\sqrt{1-\gamma}}{24} \left[15\gamma^2 - 8\gamma - 28\right]. \tag{B.1}$$

The similar calculation for the second order term gives

$$\phi^{(2)}(\gamma) = \left[B(\gamma) - 4A(\gamma)\ln\left(\frac{\gamma}{2}\right)\right]\ln\left(1 - \sqrt{1-\gamma}\right) + 2A(\gamma)\ln^2\left(1 - \sqrt{1-\gamma}\right)$$
$$+ A(\gamma)\ln^2(\gamma) - \frac{B(\gamma)}{2}\ln(\gamma) - 4A(\gamma)\operatorname{Li}_2\left(\frac{1}{2} + \frac{\sqrt{1-\gamma}}{2}\right)$$
$$- 2A(\gamma)\ln^2(2) + \frac{\pi^2}{3}A(\gamma) + C(\gamma), \tag{B.2}$$

with

$$A(\gamma) = -\left(\frac{105}{128}\right)^2 \frac{1}{32}\left(5\gamma^3 + 6\gamma^2 + 8\gamma + 16\right), \tag{B.3}$$

$$B(\gamma) = -\left(\frac{105}{128}\right)^2 \frac{1}{48}\left(33\gamma^3 - 150\gamma^2 + 96\gamma - 224\right), \tag{B.4}$$

$$C(\gamma) = \left(\frac{105}{128}\right)^2 \frac{1}{144}\sqrt{1-\gamma}\left(279\gamma^2 - 68\gamma + 524\right), \tag{B.5}$$

and the dilogarithm defined as $\operatorname{Li}_2(x) = \int_1^x dt \frac{\ln(t)}{1-t}$ [note that sometimes a different convention is used, in which the integral is defined as giving $\operatorname{Li}_2(1-x)$ instead].

In the limit $\gamma \to 0$, the above results reduce at leading order to $\phi^{(1)} \simeq -(105/128)\ln(\gamma)$ and $\phi^{(2)} \simeq (105/128)^2 \ln^2(\gamma)/2$, respectively. These expressions that can be obtained simply by keeping the leading contribution $\tilde{I}(0, \gamma')$ in the simplified kernel and taking $\phi^{(0)} = 1$ as the first term in the series.

# C Power spectral density

To calculate the power spectrum of quasiparticle number fluctuations, we follow the approach of Ref. [34]; we summarize here the necessary steps solely to make our work self-contained, with no claim of novelty. We assume short phonon lifetimes compared to the quasiparticle lifetimes to neglect fluctuations in the phonon number; the applicability of this assumption to aluminum resonators is discussed also in Ref. [17]. Using that each recombination/generation process involves two quasiparticles, we can identify their respective rates in Eq. (49), which we then use to set up a master equation for the probability $P(N, t|k, 0)$ of finding $N$ quasiparticles at time $t$ if there were $k$ quasiparticles at time 0,

$$\frac{dP(N, t|k, 0)}{dt} = -[g(N) + r(N)]P(N, t|k, 0) + g(N-2)P(N-2, t|k, 0)$$
$$+ r(N+2)P(N+2|k, 0). \tag{C.1}$$

The recombination rate – renormalized by phonon trapping – is

$$r(N) = \frac{1}{2}\tilde{R}N^2, \tag{C.2}$$

where $\tilde{R} = R/\mathcal{V}$ with $\mathcal{V}$ the resonator volume and $N = N_{qp}\mathcal{V}$ the quasiparticle number, and the generation rate is

$$g(N) = \frac{1}{2}\left(\tilde{G}_{T_B} + \tilde{G}_{\tilde{T}_B} + G(T_*/\Delta)N\right), \tag{C.3}$$

with $\tilde{G} = G\mathcal{V}$. To check the identification of the rates, we multiply both sides of the master equation by $N$ and sum over $N$ to find

$$\frac{d}{dt}\langle N(t)\rangle_k = 2\langle g(N(t))\rangle_k - 2\langle r(N(t))\rangle_k, \tag{C.4}$$

where the expectation value of a function of quasiparticle number $A(N)$ at time $u$ under the assumption of $k$ quasiparticles at 0 defined by

$$\langle A(N(u))\rangle_k = \sum_{l=0}^{\infty} A(l)P(l,u|k,0). \tag{C.5}$$

One can verify that Eq. (C.4) agrees with Eq. (49) if fluctuations are small compared to the average quasiparticle number. In the steady state, the system approaches a probability distribution for which $\langle g(N)\rangle = \langle r(N)\rangle$; for small deviations from the steady-state expectation value $N_0 = \langle N\rangle$ we can write $N = N_0 + \Delta N$ and expand the generation/recombination rates up to second order in $\Delta N$ to find

$$g(N_0) + \frac{1}{2}g''(N_0)\langle \Delta N^2\rangle = r(N_0) + \frac{1}{2}r''(N_0)\langle \Delta N^2\rangle. \tag{C.6}$$

The autocorrelation function for the number can be written in the form

$$\phi(u) = \langle N(u)N(0)\rangle = \sum_{k=0}^{\infty} kP(k,0)\langle N(u)\rangle_k, \tag{C.7}$$

with $P(k,0)$ the probability of $k$ quasiparticles at time $t = 0$. Expanding Eq. (C.4) to linear order in the (time-dependent) fluctuation $\Delta N(t)$, solving the resulting differential equation for $\langle \Delta N(t)\rangle_k$, and inserting the result into the autocorrelation function of the fluctuations we find

$$\Delta\phi(u) \equiv \langle \Delta N(u)\Delta N(0)\rangle = \langle \Delta N(0)^2\rangle e^{-t/\bar{\tau}_r}, \tag{C.8}$$

with $1/\bar{\tau}_r = 2\tilde{R}N_0 - G(T_*/\Delta)$. To calculate the variance, we multiply both sides of the master equation by $N^2$ and sum over $N$ to get

$$\frac{d}{dt}\langle N(t)^2\rangle_k = 4\langle g(N)(N+1)\rangle_k(t) - 4\langle r(N)(N-1)\rangle_k(t). \tag{C.9}$$

Considering the steady state, expanding to second order in $\Delta N$, and using Eq. (C.6) one finds the variance

$$\langle \Delta N^2\rangle = \frac{2\langle r(N)\rangle}{\tilde{R}N_0 - \frac{1}{2}G(T_*/\Delta)} \simeq 4\tilde{R}N_0^2\bar{\tau}_r. \tag{C.10}$$

If quasiparticle creation is dominated by photons or thermal phonons, one can neglect $G$, and the variance of the quasiparticle number is equal to its expectation value. In the regime in which phonon trapping dominates, we instead have $G(T_*/\Delta) = RN_0$, and the variance is twice the quasiparticle number. Fourier transforming the autocorrelation function (C.8) leads to equation (51) for the number spectral density.

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
