# Peer review of "Nonequilibrium quasiparticle distribution in superconducting resonators: effect of pair-breaking photons"

_SciPost Physics, doi:SciPost Phys. 17, 070 (2024)_

## Round 1 · Referee Report · Anonymous (Referee 1) · 2024-2-23

Strengths
-
The authors theoretically solve an important problem, which can potentially be useful for improving the quality of low temperature detectors
-
The authors rely on well established formalism and do not introduce additional fit parameters, untested assumptions etc.
Weaknesses
-
The paper is rather technical and, therefore, it is difficult to read.
-
The paper is the follow up of Ref. [17]. Therefore, the authors should clearly state what is the difference between the current manuscript and Ref. [17]
Report
The authors theoretically investigate the effect of high frequency pair-breaking photons on the quality factor and on the frequency of a superconducting resonator. They extend their previous work [17] on this subject, and obtain approximate analytical expressions for the quasiparticle distribution function in presence of both pair breaking and of low frequency photons. They show that this function may strongly deviate from equilibrium. Finally, they use this result to study the dependence of the quality factor and of the frequency of a superconducting resonator on power and on temperature. The results reasonably well agree with the experiment. Thus, the authors provide theoretical support for the mechanism of non-equilibrium quasiparticle generation by high frequency photons.
I am sure that the reported results are correct because they are based on the well established kinetic equation formalism. Nevertheless, in my opinion, the presentation of these results can be improved. Indeed, the paper is full of complicated expressions, which are difficult to follow. I understand that one cannot fully avoid this and that the kinetic equations naturally lead to that. However, it would be beneficial for the reader if the authors would explain in simple terms the physical meaning of obtained results, and would discuss the general system setup. For example, the following issues can be additionally clarified:
-
The set of peaks in the distribution function plotted in Fig. 5 are located at energies m\omega_0. Can one interpret this as a result of multi-photon processes, at least at the qualitative level? The same question applies to Fig. 6.
-
Another unclear issue is related to the very low values of the distribution function in Figs. 5 and 6. Namely, does it matter if this function has a peak with the height 10^{-18}? What are the minimal values of f(E), which are relevant for the experiment? Do the peaks in Figs. 5 and 6 affect the quality factor in Figs. 8 and 9?
-
Why do the authors consider only one high frequency mode with the frequency \omega_0 > 2\Delta? Does it correspond to the experimental setup? I think assume that, for example, a high quality factor niobium cavity of a centimeter size should have dense spectrum around 100 GHz frequency, which corresponds to 2\Delta.
Finally, the authors should also clearly state the difference between the current manuscript and Ref. [17], which has a lot of similarities with the manuscript.
In conclusion, I recommend accepting the paper after minor changes in the presentation, which have been mentioned above.
Requested changes
The requested changes are also mentioned in the report.
-
Clearly state what is new as compared to Ref. [17]
-
Add a bit more qualitative discussion of the results

---

## Round 1 · Referee Report · Anonymous (Referee 2) · 2024-6-14

Strengths
1- Quasiparticles are a widespread nuisance in superconducting circuits, but their origin and modelling is still poorly understood, so this work appears timely 2- The work combines numerical solution of rate equations, with simpler analytical insights that could be useful for experimentalists
Weaknesses
1- The manuscript will appeal to specialists 2- It is often cryptic, and involves lots of experimental parameters 3- There is strong overlap with a previous publications by the authors (Ref [17])
Report
and technological importance, of the origin of quasiparticle poisoning
in superconducting circuits. The main emphasis is put on the
role of high energy photons, with a combination of analytical
and numerical analysis.
The paper will certainly be useful to theorists and experimentalists
in the field, but is hard to read for non specialists.
I am suggesting below several recommendations to the authors that could improve the readability of their manuscript, as well as asking some needed clarification.
Overall, this is a serious and useful study. It seems that it does not meet
any of the four main "novelty" criteria for SciPost Physics, but it would be
perfectly acceptable as SciPost Core, after revision.
Requested changes
Page 1: the highlighted (in upper case) part of the following sentence is confusing: "a large number of (non-pair-breaking) photons can “heat up” the quasiparticles by pushing them to higher energy as compared to the phonon temperature; these quasiparticles can relax by emitting phonons, so that the latter are also driven out of equilibrium. If the emission process involves a RELATIVELY LARGE NUMBER OF PAIR-BREAKING PHONONS with frequency ω > 2∆". Owing to energy conservation, I do not see how low energy photons can create enough quasiparticles as to emit a large number of pair-breaking phonons above the gap.
Page 1: in this sentence "we present the kinetic equation that determines the quasiparticle distribution; it extends the previously used kinetic equations describing the interaction of quasiparticles with photons of energy below the pair breaking threshold 2∆ [17, 23, 24] by including a contribution from a mode of energy above the threshold" Why is the assumption of a single mode justified? Typically, the photons experienced by a superconducting device will have gone through a series of complex filtering, but I don't see a reason why it should be spectrally sharp
Page 2, Eq. (2): Why is \bar{n} not a function of \omega_0 here? Why is this not an integral over the the low-energy photon density of states? Identify the various terms here w.r.t. the bare processes in Fig. 1
Page 2, Eq. (4) and Eq. (5): Why is \bar{n}{PB} not a function of \omega here? Identify the various terms here w.r.t. the bare processes in Fig. 1
Page 2, before Eq. (4): State clearly that pair breaking photons have a definite energy omega_{PB}
Page 2, Fig.1: 1/ why is the diagram for quasiparticle to photon recombination omitted? 2/ are the higher order diagrams g) and h) really useful in the analysis?
Page 3, above Eq. (6): I understand that Pauli blocking factors can be replaced to unity at low temperature, but: 1/ is that really legitimate when the non-equilibrium distribution function is not known before hand? 2/ doesn't that violate detailed balance or other sum rules?
Page 3, Eq. (6): 1/ if the integral over \epsilon starts at zero, why is the BCS term diverging at \epsilon+\omega=0 and not \epsilon+\omega=\Delta? 2/ St_g^{hon} is not defined 3/ why doesn't the phonon population $n$ enter here explicitely?
Page 3: Globally, I find this whole section hard to read
Page 3, Eq. (12): going from the first line to the second seems unjustified, since the factor U(E) has a divergence at the superconducting gap. Also, why can it be replaced by unity?
Page 4-5: the mapping and analysis of the Volterra equation is relatively well-explained. Can the author clarify the physical difference between \gamma_\star and \gamma_\star' ?
Page 6, Fig. 4 (top plot): Indicate "exact" as a black line inside the figure
Page 7: it would be nice to have a few sentences at the end of this section that summarizes the main results. In particular, regarding the range of maximal values for f(\gamma) in Fig. 4, what do they imply experimentally? Is 10^(-5) a "big" or a "small" number?
Page 10: It is not clear where Eq. (49) comes from
Page 11, Eq. (54-55) and Fig 8: 1/ Can the extrinsic quality factor Q_{i,ext} be extracted from this fit, and what could one learn from it? For instance, if it is due to dielectric losses, its temperature and frequency dependence is known in the literature 2/ It seems that Fig. 10 fits better without Q_{i,ext}, which does not make sense. Can the authors clarify? 3/ why don't the properties (e.g. impedance) of the considered resonator enter the equation?
Page 13, Eq. (59): can the authors explain briefly where the equation for the energy shift comes from, is it just Kramers-Kronig?
Recommendation
Ask for minor revision

---

## Round 2 · Referee Report · Anonymous (Referee 2) · 2024-8-7

Report

The authors have addressed in detail my comments for clarification.
I recommend publication of the manuscript

Recommendation

Publish (easily meets expectations and criteria for this Journal; among top 50%)

---

## Round 2 · Author Response

Dear Editor,

we submit our revised manuscript for your further consideration. We include below detailed responses to the Referees' reports, which contain descriptions of the changes made to the article.

Response to Report 1:

We thank the Referee for carefully reading the manuscript, recommending its acceptance, and suggesting improvements in the presentation. In the revised manuscript, we have added a new Fig.1 that represent the main system setup we investigate and mention more explicitly what distinguish the present work from the previous one, for instance in the captions to Figs. 1 and 2 and at the beginning of Sec.III. More discussions of our results have been added, for example at the end of Sec.III and in Sec.V.

Regarding the three specific points raised in the report, we have addressed them as follows:

1. We have clarified the processes included in the photon collision integral and the origin of the multiple peaks in the text after Eq.(39)

2. We now state clearly that the “second peak” of magnitude 10^{-18} in Fig. 7 (former Fig.6) is not relevant, see text after Eq.(48).
The experimentally relevant order of magnitude for the value of the distribution function near the gap is discussed at the end of Sec.III by relating it to the normalized quasiparticle density x_qp.
The details about the peaks are not relevant to estimate the quality factor, see text in brackets after Eq.(55).

3. We agree with the Referee that the assumption of a single high-frequency mode can be an idealized one in some cases, such as bulk cavities. The experimental data we analyze is for a coplanar waveguide resonator; in 2D chips, one or a few spurious modes could dominate the high-frequency response. Also, if the noise source is indeed the microwave generator, the high frequency photons could be emitted with sufficient rate only at one or a few frequencies. Therefore, the assumption is not unrealistic. In any case, parts of our calculations are straightforward to generalize for multiple modes. We had already pointed this out briefly at the very end of Sec.IV, we have now added mentions of such possible extensions also in the text preceding subsection III.A and that preceding subsection V.A

Response to Report 2:

We thank the Referee for finding our work “serious and useful” and for making several recommendations for improvements. In our view, the manuscript satisfies at least one of the “novelty” criteria to be accepted in SciPost Physics: the issue of the low-temperature saturation of the quality factor in superconducting devices is a long-standing problem, and our results point to the significant impact that a small number of pair-breaking photons can have on this quantity, an effect that was not previously appreciated. In the first version of the article, this was perhaps not made sufficiently clear. In the revised version, we now emphasize that the experimental data of Ref. 18 cannot be explained by standard approaches, such as losses due to two-level systems, while our approach is able to fit the data using a minimal number of additional parameters.

We specify below the changes implemented to address the detailed questions of the Referee. Together with other improvements (for instance, the addition of the new Fig.1), we believe that the revised manuscript is appropriate for publication in SciPost Physics.

Page 1: We have expanded the introduction to clarify the heating mechanism by which low-energy photons lead to the creation of additional quasiparticles. The Referee is correct in saying that this mechanism is not very efficient: while it can eventually lead to a saturation of the quality factor, the predicted value for the latter is much higher than what experimentally measured, as we now state. This was in fact a key finding of our previous work, Ref.17: low-energy photons alone cannot explain the data.

Page 1 (now page 2): The use of a single mode could be justified in some situations – this could be for instance a spurious mode in a planar chip, or noise generated at a specific frequency by control electronics. We now mention how the approach can be extended to multiple modes, see for example the text after Eq.(3), that before Sec. III.A, the end of Sec.IV (already in the first version), and in particular the text before Sec. V.A, where we explain why the assumption does not qualitatively affect our results.

Page 2, Eq.(2): The reason why a single non-pair-breaking photon is considered is now explained in the text after Eq.(3). Throughout the text, we related terms in the collision integrals to the diagrams in Fig.2 (former Fig.1).

Page 2, Eq.(4) and (5) and Page 2, before Eq.(4): These points are addressed in the previous two points.

Page 2, Fig. 1 (now Fig.2): As explained in Sec. III.A, photon assisted recombination is negligible compared to phonon assisted recombination. For added clarity, this is now mentioned in the caption of Fig.2 as well.
Diagram g) leads to a renormalization of the recombination coefficient, we have added reference to this diagram after Eq.(8).
Diagram h) affects the density through the G(T_*/Δ) term and is only important for very large photon numbers. This is now mentioned in the caption of Fig 2 and we refer to diagram h) before Eq.(50).

Page 3, above Eq. (6): The consistency of the f<<1 assumption should in fact be verified once the distribution functions has been calculated, as we now state in the text before Eq.(6). For the regime of low quasiparticle densities we consider, this assumption is in general valid.
When making approximations it is important that quasiparticle number conserving terms remain number conserving; this is the case when neglecting Pauli blocking.

Page 3, Eq.(6): The notation for energy \epsilon is now more clearly defined after Eq.(6): \epsilon=0 correspond to the BCS gap energy E=\Delta. We have also added the definition of St_g^{Phon}. The phonon occupation does not appear because, as mentioned before Eq. (6), the phonon temperature is assumed to be sufficiently low to permit ignoring phonon absorption by quasiparticles; this assumption is discussed in more detail in Sec. III.A.

Page 3: We have improved section two by: adding a new Fig.1; relating the diagrams in Fig.2 (former Fig.1) to the collision integral; clarifying the notation and definitions of various terms.

Page 3, Eq.(12): The function U(E_1,E_2) has a divergence only for E_2 approaching \Delta while is regular for E_1 near \Delta, see the definition in the text after Eq.(3). Since E+\omega_PB > 3\Delta, we approximate U with the value it takes for large second argument.

Page 4-5: We now discuss the difference between \gamma’_* and \gamma_* in the text after Eq.(28)

Page 6, Fig.4 (now Fig. 5): We denote the black line with the label “numeric”

Page 7: We briefly discuss the results of the section and comment on the experimentally relevant values for f at the end of Sec.III.

Page 10: We briefly describe the derivation of Eq.(49) in the text after Eq.(50)

Page 11, Eq.(54-55): The values of Q_{I,ext} are those shown in Fig.10 (former Fig.9), see also text after Eq.(55). Their overall decrease with increasing power rules out an explanation in terms of dielectric losses due to two-level systems; we now make this important point in the introduction as well as in Sec.V.
The fitting in Fig.11 (former Fig.10) is fully based on our theory, without any additional extrinsic mechanisms, and requires fewer parameters than a phenomenological fit. We remark on this in the paragraph after that containing Eq.(56).
The properties of the resonator enter implicitly into the problem via the photon frequency \omega_0 and the coupling constant c^{QP}_{Phot} (for the latter, see Appendix A, Ref.17 and references there).

Page 13, Eq.(59): We briefly discuss the origin of the equation in the text preceding it. The relation to Kramers-Kronig is mentioned in the conclusions.

---

## Editorial Decision

published